# A metabolic interplay coordinated by HLX regulates myeloid differentiation and AML through partly overlapping pathways

Indre Piragyte[1,2], Thomas Clapes [1], Aikaterini Polyzou[1,2], Ramon I. Klein Geltink[3], Stylianos Lefkopoulos [1,2], Na Yin[1], Pierre Cauchy[1], Jonathan D. Curtis[3], Lhéanna Klaeylé[1], Xavier Langa [4], Cora C.A. Beckmann[5], Marcin W. Wlodarski [5], Patrick Müller [6], Dominic Van Essen[7], Angelika Rambold[8,9], Friedrich G. Kapp[5], Marina Mione [10], Joerg M. Buescher [3], Erika L. Pearce[3], Alexander Polyzos [11] & Eirini Trompouki [1]

The H2.0-like homeobox transcription factor (HLX) regulates hematopoietic differentiation and is overexpressed in Acute Myeloid Leukemia (AML), but the mechanisms underlying these functions remain unclear. We demonstrate here that HLX overexpression leads to a myeloid differentiation block both in zebrafish and human hematopoietic stem and progenitor cells (HSPCs). We show that HLX overexpression leads to downregulation of genes encoding electron transport chain (ETC) components and upregulation of PPARδ gene expression in zebrafish and human HSPCs. HLX overexpression also results in AMPK activation. Pharmacological modulation of PPARδ signaling relieves the HLX-induced myeloid differentiation block and rescues HSPC loss upon *HLX* knockdown but it has no effect on AML cell lines. In contrast, AMPK inhibition results in reduced viability of AML cell lines, but minimally affects myeloid progenitors. This newly described role of HLX in regulating the metabolic state of hematopoietic cells may have important therapeutic implications.

[1] Department of Cellular and Molecular Immunology, Max Planck Institute of Immunobiology and Epigenetics, 51 Stübeweg, 79108 Freiburg, Germany. [2] Faculty of Biology, University of Freiburg, Schänzlestraße 1, 79104 Freiburg, Germany. [3] Department of Immunometabolism, Max Planck Institute of Immunobiology and Epigenetics, 51 Stübeweg, 79108 Freiburg, Germany. [4] Institute of Anatomy, University of Bern, Baltzerstrasse 2, 3012 Bern, Switzerland. [5] Division of Pediatric Hematology and Oncology, Department of Pediatrics and Adolescent Medicine, Medical Center - University of Freiburg, Faculty of Medicine, University of Freiburg, Mathildenstr. 1, 79106 Freiburg, Germany. [6] Systems Biology of Development Group, Friedrich Miescher Laboratory of the Max Planck Society, Max-Planck-Ring 9, 72076 Tübingen, Germany. [7] Institute for Research on Cancer and Aging Nice, 28 Ave de Valombrose, 06107 Nice Cedex 02, France. [8] Department of Developmental Immunology, Max Planck Institute of Immunobiology and Epigenetics, 51 Stübeweg, 79108 Freiburg, Germany. [9] Center for Chronic Immunodeficiency, Freiburg University Medical Center, 55 Hugstetter Street, 79106 Freiburg, Germany. [10] Centre for Integrative Biology, University of Trento, Via Sommarive, 9, 38123 Povo Trento, Italy. [11] Biomedical Research Foundation of the Academy of Athens, 4 Soranou Ephessiou Street, 115 27 Athens, Greece. These authors contributed equally: Indre Piragyte, Thomas Clapes, Aikaterini Polyzou. Correspondence and requests for materials should be addressed to E.T. (email: trompouki@ie-freiburg.mpg.de)

Long-term hematopoietic stem cells (LT-HSCs) are multipotent cells with self-renewal capacity primarily responsible for replenishing the entire hematopoietic system[1–7]. LT-HSC differentiation into mature blood and immune cells is a tightly regulated and multifaceted process. Transcription factors govern the mechanisms that maintain the balance between LT-HSC differentiation and self-renewal, or stemness[8–10], and any perturbation in this process can ultimately lead to disease.

While it is well established that homeobox (HOX) transcription factors play a central role in hematopoietic development and disease, less is known about the function of non-clustered HOX factors in the hematopoietic system[11,12]. The non-clustered H2.0-like homeobox transcription factor (HLX) has been recently identified as an important regulator of hematopoiesis. During development, HLX deficiency leads to a decrease in the colony-forming capacity of fetal liver cells[13–16], and in adult hematopoiesis HLX regulates Th1/Th2 differentiation during T-cell development[17–20]. Recent evidence shows that HLX is essential for HSC maintenance and self-renewal[21–23]. Increased expression of HLX compromises self-renewal and eventually results in a myelomonocytic differentiation block concomitant with aberrant proliferation of myeloid progenitors[21]. Mechanistically, it has been suggested that this function of HLX in HSC maintenance and self-renewal is mediated by the p21-activated kinase PAK1. Indeed, it was demonstrated that inhibition of HLX or PAK1 induces differentiation and apoptosis of AML cells[21,22]. Consistent with this phenotype, HLX is overexpressed in 87% of AML patients and those presenting higher HLX expression have lower survival rates[21]. Recently, HLX has been shown to play a role in the browning of white adipose tissue, suggesting that this transcription factor is involved in the metabolic control of cell differentiation[24].

Despite the pleiotropic functions of HLX and its critical regulatory role in multiple processes, particularly in hematopoiesis, only few direct downstream targets have been identified. Moreover, mechanistic insights into the function of HLX in hematopoiesis and myeloid differentiation are lacking. Thus, understanding the physiological roles of HLX in hematopoietic development and disease, including leukemia, remains a central issue in HSC biology.

Here, we use zebrafish, human hematopoietic stem and progenitor cells (HSPCs), and AML cell lines to explore the underlying mechanisms of HLX function during hematopoiesis. We show that HLX overexpression results in an aberrant proliferation of HSPCs and a myeloid differentiation block in both systems. We find that HLX exerts its biological function in hematopoiesis, at least in part, by direct control of electron transport chain (ETC) and PPARδ gene expression. Metabolic stress leads to an elevation of AMP-activated kinase (AMPK) levels and autophagy. Modulation of PPARδ signaling can rescue the hematopoietic phenotypes of HLX in both zebrafish and human cells, but has no obvious impact on AML cells. In contrast, AMPK inhibition reduces viability of AML cell lines, but minimally affects primary cells. This newly discovered link between HLX and metabolism could be a promising new avenue for treating hematological diseases.

## Results

### HLX overexpression blocks zebrafish myeloid cell maturation.
To investigate the mechanisms underlying the role of HLX in promoting AML, we examined hematopoiesis in HLX-overexpressing zebrafish models. We crossed the Tg(fli1a: Gal4FF)ubs3[25] line with our Tg(UAS:HLX-GFP) to induce expression of human HLX (hHLX) in endothelial and hematopoietic cells and named these fish fli:hHLXOE. We chose to use human HLX in an effort to demonstrate conservation and translate our results into the human gene function. fli:hHLX overexpression led to increased specification of HSPCs at 36 h post fertilization (hpf) in the Aorta–Gonad–Mesonephros region as shown by runx1 whole-mount in situ hybridization (WISH) (Fig. 1a and Supplementary Fig. 1a). The increased number of HSPCs led to increased rag1 staining in the thymus at 96 hpf (Fig. 1b). WISH for the early myeloid marker pu.1 revealed that these transgenic fish presented an expansion of myeloid progenitors (Fig. 1c). We then asked whether HLX overexpression affects myeloid cell maturation. May–Grünwald–Giemsa staining showed that fli:hHLXOE embryos have a significantly larger proportion of immature myeloid cells (75.5%) when compared to their wild-type counterparts (35.3%) at 48 hpf (Fig. 1d). EdU staining revealed hyperproliferation of endothelial cells, which may be the underlying cause of the increased number of HSPCs (Fig. 1e). This enhanced proliferation does not induce apoptosis in fli:hHLXOE embryos, as shown by TUNEL assay (Supplementary Fig. 1b).

At 48 hpf most of the myeloid cells are derived from primitive/prodefinitive and not definitive hematopoiesis. To verify that the differentiation block occurs in myeloid cells that arise from HSPCs we crossed Tg(Mmu.Runx1:GAL4) fish to Tg(UAS:HLX-GFP) and named the progeny Runx:hHLXOE. These fish express hHLX only in HSPCs. In this model we show that more HSPCs are specified at 26 hpf as indicated by runx1 staining, followed by modestly elevated c-myb staining and mRNA expression at 3 dpf (days post fertilization) (Supplementary Fig. 1c, d). We then verified by May–Grünwald–Giemsa staining that at 5 dpf HLX overexpression in HSPCs leads to a strong myeloid differentiation block without affecting erythrocyte numbers (Fig. 1f). This result was verified by qPCR and WISH for a panel of mature myeloid markers and gata1 as a marker of erythroid differentiation (Supplementary Fig. 1d, e).

Together, these results suggest that HLX overexpression results in increased numbers of HSPCs and blocks myeloid cell differentiation.

### HLX is required for HSPC formation.
To examine hematopoiesis in hlx1 knockdown animals, we generated hlx1 morphants (hlx1MO) using a previously published translational morpholino[26]. Inhibition of hlx1 translation in zebrafish embryos decreased the pool of HSPCs, as shown by runx1 and c-myb WISH at 36 hpf, respectively (Fig. 1g and Supplementary Fig. 1a, f). To quantify the number of HSPCs in hlx1MO animals, we injected either the translational morpholino used in all experiments, or a splicing morpholino[26] in Tg(Runx:mCherry) fish[27], a line with fluorescent HSPCs. The number of m-Cherry[+] cells (HSPCs) was significantly decreased in both types of morphants (Supplementary Fig. 1g). WISH for rag1 showed that hlx1MO have fewer thymocytes at 96 hpf (Fig. 1g), when compared to the control embryos. To exclude the possibility that HSPC loss is due to arterial or vascular endothelial defects, we analyzed the expression of arterial (ephrinB2α) and endothelial (kdrl) markers in hlx1MO. In agreement with previous reports, deregulation of hlx1 affects the identity of stack cells, but does not seem to have severe effects on the cardinal vein (Supplementary Fig. 1h)[26]. Additionally, TUNEL assays demonstrated that HSPC loss in hlx1MO is not caused by apoptosis, whereas EdU staining showed reduced proliferation of endothelial cells (Supplementary Fig. 1b, and Fig. 1e, respectively).

We then attempted to rescue the morphant phenotype by overexpressing hHLX in endothelial/hematopoietic (fli:hHLX) or hemogenic endothelial cells (Runx:hHLX). Remarkably, hHLX

overexpression in both cell types rescued HSPC loss (Fig. 1h and Supplementary Fig. 1i).

Collectively, these data show that HLX regulates the formation of HSPCs.

**HLX regulates genes involved in metabolism.** To understand the mechanisms of HLX function in hematopoiesis, we performed RNA-Seq on FACS-sorted endothelial cells from *fli*:h*HLX*OE (*fli:kaede*) and *hlx1*MO (*kdrl:GFP*) embryos at 48 hpf. Compared with control embryos, we identified 2950 downregulated and

3419 upregulated genes that changed by more than two-fold (negative binomial test (NBT), $P < 0.05$) in *fli*:h*HLX*OE embryos (Fig. 2a, Supplementary Fig. 2a, and Supplementary Data 1). On the other hand, 942 genes were downregulated and 1162 genes upregulated over two-fold (NBT, $P < 0.05$) in *hlx1*MO animals (Fig. 2a, Supplementary Fig. 2a, and Supplementary Data 1). Seventy-nine (8%, hypergeometric test (hg.t), $P < 0.01$) of the downregulated genes in *hlx1*MO were inversely correlated in *fli*:h*HLX*OE. In total 869 genes were deregulated in both *hlx1*MO and *fli*:h*HLX*OE (hg.t, $P < 2.27\text{E-}120$). Next, to identify pathways

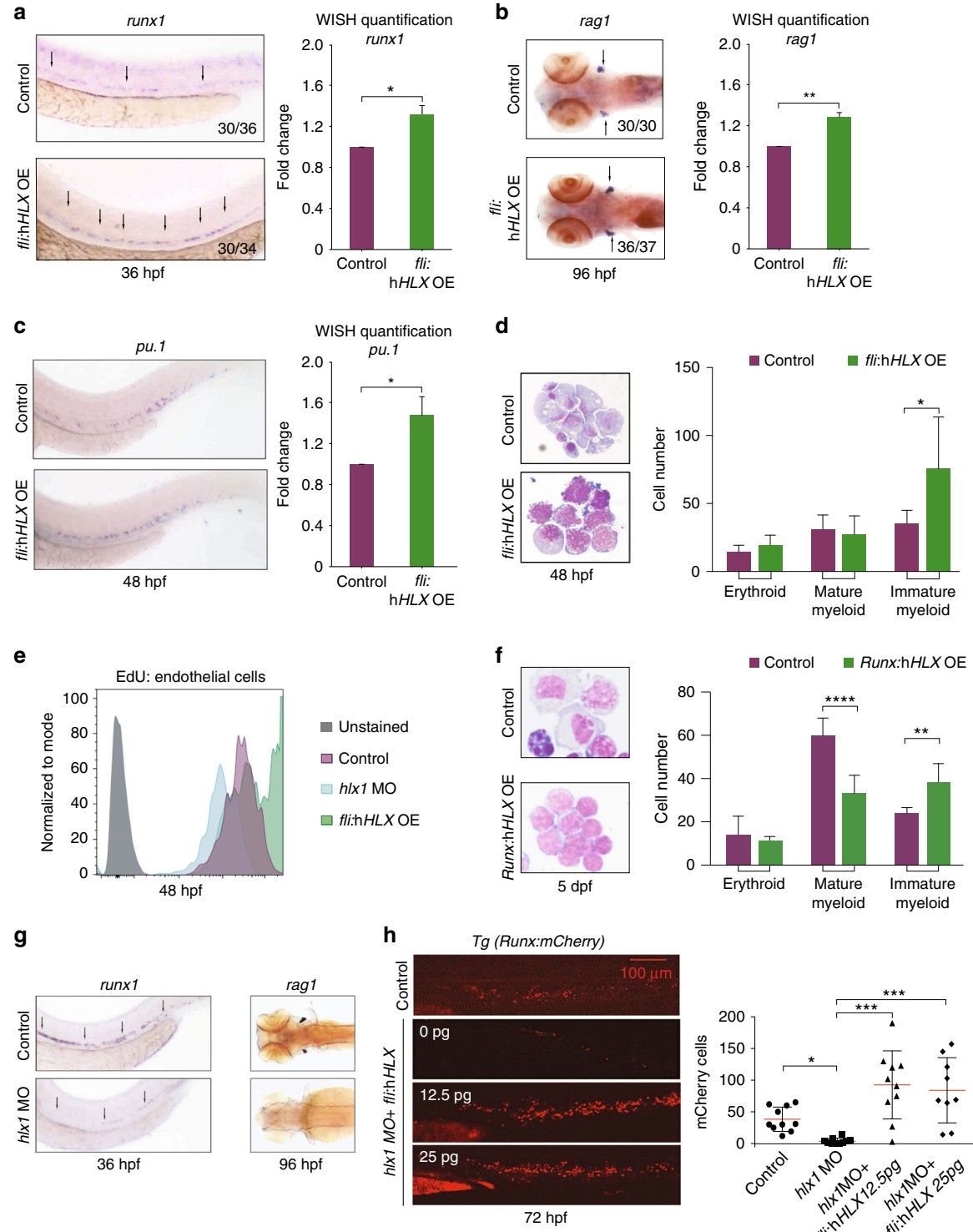

deregulated upon h*HLX* overexpression or *hlx1* knockdown, we performed ingenuity pathway analysis (IPA), gene ontology, and gene set enrichment analysis (GSEA) and created networks using Cytoscape (Fig. 2b, Supplementary Fig. 2b, c, and Supplementary Data 1). These analyses showed that HLX is a pleiotropic transcription factor that regulates fundamental processes. Interestingly, the two canonical pathways most affected by h*HLX* overexpression were oxidative phosphorylation (OXPHOS) (right-tailed Fisher Exact Test (rtFET), $P < 3.16E-23$) and mitochondrial dysfunction (rtFET, $P < 1.99E-21$, Fig. 2b and Supplementary Data 1). GSEA also indicated that genes downregulated in *fli*:h*HLX*OE or deregulated in *hlx1*MO are associated with changes in mediators of OXPHOS (Supplementary Fig. 2c). Multiple genes of the mitochondrial ETC belonged to the above-mentioned categories and were downregulated in *fli*:h*HLX*OE embryos (Supplementary Data 1). Gene deregulation detected by RNA-Seq was confirmed by qPCR in *fli*:h*HLX*OE and *Runx*:h*HLX*OE or *hlx1*MO embryos (Supplementary Fig. 2d–f). These results suggest that HLX regulates mitochondrial metabolic genes. This unexpected finding has important implications, as mitochondrial metabolism is essential for LT-HSCs stemness[28,29] and AML patients can present defects in mitochondrial metabolism[30].

Consistent with the transcriptional deregulation observed in the *hlx1*MO, we also detected differences in chromatin accessibility, by performing ATAC-Seq in endothelial (*kdrl*:*GFP*) cells sorted from control and *hlx1*MO embryos at 48 hpf. Using the MACS2 (version 2.1.0) bdgdiff command with the default settings we identified 16,409 peaks that were either lost or gained in *hlx1*MO, when compared to wild-type siblings (Fig. 2c and Supplementary Data 2). Using Genomic Regions Enrichment of Annotations Tool (GREAT) analysis, we found that the majority of peaks were located between 5 and 500 kB of the transcription start site (TSS) and regulate a variety of processes (Supplementary Fig. 3a and Supplementary Data 2). We used a nominal cut-off of 25 kb from the TSS and assigned the differential peaks to 6431 genes (Supplementary Data 2). IPA analysis of these genes revealed that mitochondrial dysfunction (rtFET, $P < 3.47E-09$) and OXPHOS (rtFET, $P < 8.13E-08$) were among the upper enriched categories (Supplementary Data 2). To gain mechanistic insights we performed footprinting analysis[31] in control and *hlx1*MO datasets and obtained 54,588 and 50,471 footprints, respectively. Motif discovery revealed HOX motifs enriched in control-only footprints and AP-1 motifs in *hlx1*MO-only footptrints (Supplementary Fig. 3b). To determine whether these motifs were significantly differentially footprinted between the two datasets, we computed motif self-enrichments and co-occurrence enrichments from specific footprint populations over background occurrences computed from reciprocal datasets. This analysis revealed loss of HOX motifs and gain of AP-1 motifs in *hlx1*MO (co-occurrence enrichment computation $z = 3.583$ and $z = 13.241$, respectively) (Supplementary Fig. 3c). Tn5 insertion

profiles at these sites revealed diverging profiles at footprinted Hoxc9 and AP1 motifs (Student's *t*-test, Hoxc9 control specific $P = 2.9884E-34$, AP-1 *hlx1*MO specific $P = 1.24262E-37$) (Fig. 2d). Analysis of relative footprint occurrences to their cognate datasets also revealed that Hoxc9 and AP-1 motifs were more present in control and *hlx1*MO-specific footprints, respectively (Supplementary Fig. 3d).

After integration of our RNA-Seq and ATAC-Seq data from the *hlx1*MO, we identified 739 (35%, hg.t., $P < 1.930E-62$) deregulated genes with changes in chromatin accessibility (Fig. 2e and Supplementary Data 2). Comparison of the ATAC-Seq from *hlx1*MO and RNA-Seq from *fli*:h*HLX*OE embryos demonstrated that 1963 (31%, hg.t., $P < 3.082e-116$) deregulated genes exhibited differences in chromatin accessibility (Fig. 2e, Supplementary Fig. 3e, and Supplementary Data 2). Interestingly, ETC genes that were downregulated upon h*HLX* overexpression also showed changes in chromatin accessibility in two different types of ATAC-Seq analyses (whole region or sub-nucleosomal) (Fig. 2f and Supplementary Data 2). Concomitantly, some ETC genes showed upregulation of expression in *hlx1*MO (Fig. 2f).

Finally, *ppardb* (peroxisome-proliferator activated receptor δ) also showed changes in chromatin accessibility in sub-nucleosomal analysis (Fig. 2e and Supplementary Data 2). Recently PPARδ, a regulator of metabolic functions, was shown to be critical for LT-HSC stemness[32]. We therefore performed qPCR for *ppar* genes in *hxl1*MO, *fli*:h*HLX*OE, and *Runx*:h*HLX*OE embryos. Indeed, the expression of the *ppar* receptors was significantly decreased in *hxl1*MO (Fig. 2g), but only *pparda* and *ppardb* were increased in *fli*:h*HLX*OE or *Runx*:h*HLX*OE embryos (Fig. 2h, i).

Together, these data demonstrate that Hlx1 regulates the expression of ETC and *ppar* genes in zebrafish endothelial cells and HSPCs.

### PPARδ signaling rescues zebrafish hematopoietic phenotypes.

The results above suggest that HLX regulates genes involved in metabolism. To investigate this further, we asked whether HLX modulation has any functional consequences for cell metabolism in vivo. We assessed OXPHOS by seahorse metabolic flux analysis, which measures oxygen consumption rate (OCR). These experiments revealed that there is a reduction in spare respiratory capacity in *fli*:h*HLX*OE embryos (Fig. 3a). Given the transcriptional deregulation of ETC genes upon h*HLX* overexpression, we measured mitochondrial membrane potential by using tetramethylrhodamine ethyl ester perchlorate (TMRM). TMRM is a dye that is sequestered in active mitochondria, and reflects the ability of cells to produce ATP. As expected, although the total mitochondrial mass was unchanged in *fli*:h*HLX*OE embryos, TMRM was significantly lower (Fig. 3b). In contrast, TMRM was higher in endothelial cells of *hlx1*MO, yet the total mitochondrial

---

**Fig. 1** *hlx1* regulates hematopoietic stem cell formation and myeloid cell maturation in zebrafish. **a–b** Whole-mount in situ hybridization (WISH) for *runx1* (**a**) and *rag1* (**b**) in control or *fli*:h*HLX*OE zebrafish embryos at 36 or 96 hpf, respectively. Arrows indicate HSPCs. Numbers in the bottom right corner of panels indicate the number of zebrafish embryos with the indicated phenotype compared to the total number of zebrafish analyzed. Quantification of WISH was performed using FIJI software and statistical significance of three independent experiments in 12 zebrafish embryos was evaluated by Student's *t*-test, *$P < 0.05$, **$P < 0.01$ (mean + s.d.). **c** WISH for the early myeloid marker *pu.1* in control or *fli*:h*HLX*OE zebrafish embryos at 48 hpf. Numbers and WISH quantification was performed as described above ($n = 3$, in total 12 fish, Student's *t*-test, *$P < 0.05$, mean + s.d.). **d** Zebrafish caudal hematopoietic tissue (CHT) smears stained with May–Grünwald–Giemsa stain in control or *fli*:h*HLX*OE embryos at 48 hpf. On the right, cell number counts of the indicated cell populations from 10 fish ($n = 3$, mean + s.d., Student's *t*-test, *$P < 0.05$). **e** EDU assay at 48 hpf in control, *hlx1* morphants (*hlx1*MO) or *fli*:h*HLX*OE zebrafish cells ($n = 2$). **f** Zebrafish CHT smears stained with May–Grünwald–Giemsa stain in control or *Runx*:h*HLX*OE embryos at 5 dpf. On the right, cell number counts of the indicated cell populations from 10 fishes, in two independent experiments (mean + s.d., Student's *t*-test, **$P < 0.01$, ****$P < 0.0001$). **g** WISH for *runx1* and *rag1* in control or *hlx1*MO embryos at 36 and 96 hpf, respectively. **h** Representative images of *Tg(Runx:mCherry)* embryos at 72 hpf, injected or not with *hlx1* morpholino and the indicated amounts of *fli*:h*HLX* construct. Numbers of mCherry-positive HSPCs from each embryo are represented in the graph ($n = 10$; mean + s.d., Student's *t*-test, *$P < 0.05$, ***$P < 0.001$)

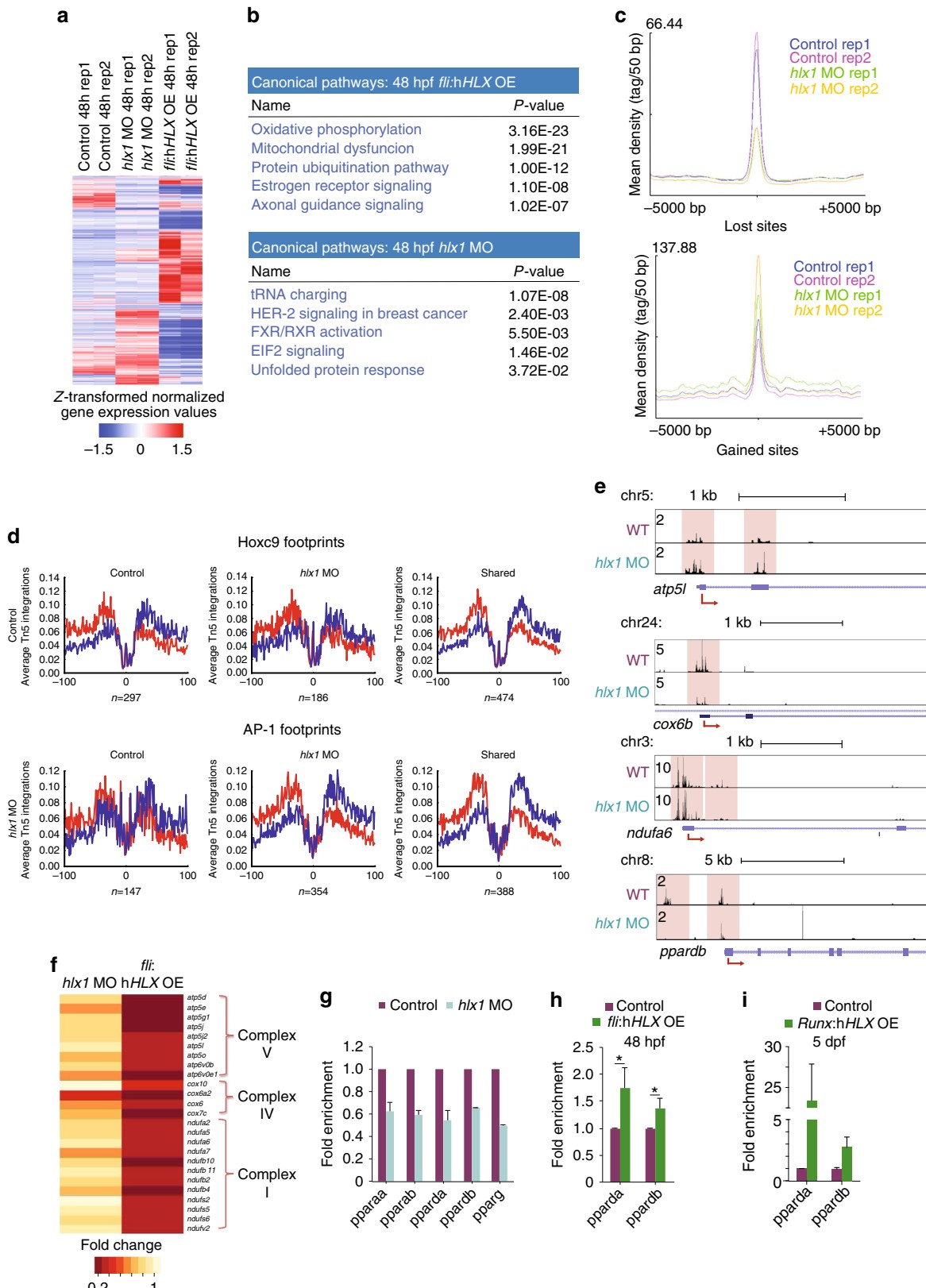

mass was slightly decreased (Fig. 3c). These data demonstrate that *HLX* overexpression affects mediators of OXPHOS and mitochondrial membrane potential in zebrafish in vivo.

As HLX regulates *PPARδ* gene transcription, a gene essential for lipid metabolism[33], and in some systems mitochondrial biogenesis[34], we assessed whether pharmacological modulation of PPARδ can rescue defects in mitochondrial membrane potential maintenance. Indeed, we were able to rescue TMRM levels by using a specific PPARδ antagonist (GSK3787) in *fli*:h*HLX*OE embryos, accompanied by an increase in the mitochondrial mass

(Fig. 3d). This PPARδ antagonist also elevated the TMRM of the control cells, albeit to a lower extent. We then attempted to rescue the HLX-induced hematopoietic phenotypes by modulating the activity of PPARδ. First, we treated fli:hHLXOE zebrafish embryos with a PPARδ antagonist (GSK3787) and examined erythro-myeloid cells at 48 hpf. Pharmacological inhibition of PPARδ significantly reduced the number of immature myeloid cells and partially rescued the myeloid block (Fig. 3e). Conversely, a PPARδ agonist (L165,041) rescued HSPC loss in hlx1MO embryos (Fig. 3f).

These results demonstrate that modulation of PPARδ rescues HLX-induced defects in HSPC formation and myeloid cell maturation.

**HLX regulates ETC and PPAR genes in human cells.** We next addressed the physiological relevance of the interplay between HLX and metabolic regulation in humans. From the database BloodSpot[35] and previously published data[36,37], we confirmed that HLX is expressed in murine and human hematopoietic progenitor cells, but also in mature myeloid lineages (Supplementary Fig. 4a–c). Moreover, HLX is upregulated in AML patient samples, as it has been previously published[21] and according to information from cBioportal (Supplementary Fig. 4d). We next investigated whether AML patients show similar transcriptional signatures to those of hHLXOE zebrafish embryos. We obtained data generated by the TCGA Research Network (http://cancergenome.nih.gov/) from 193 AML patients. Normalized gene expression data of 156 AML samples were used for further analysis. We performed correlation analysis among all genes with significant expression (RSEM values >50) across all patients and the HLX gene. We then did GO and IPA analyses (Supplementary Data 3) in the HLX positively (Pearson Correlation Coefficient >0.4, 824 genes) and negatively (Pearson Correlation Coefficient <−0.4, 542 genes) correlated genes. Consistent with the zebrafish data, OXPHOS (rtFET, $P < 0.0003$), mitochondrial dysfunction (rtFET, $P < 0.0107$), and other metabolic categories were among the pathways that showed positive correlation with HLX expression in AML patients in IPA analysis (Supplementary Fig. 4e and Data 3). Interestingly, HLX expression in patients positively correlates with PPARδ expression (Pearson Correlation Coefficient 0.42) (Supplementary Data 3).

These genomic analyses in AML patients suggest that the role of HLX in metabolic regulation is conserved in humans. To identify genes that are directly regulated by HLX in human hematopoietic cells, we performed ChIP-Seq in two mammalian cell lines (chronic myelogenous leukemia CML: K562 and acute myeloid leukemia-AML HL60) overexpressing a FLAG-tagged version of hHLX (Supplementary Fig. 5a). In both cell lines, the majority of HLX ChIP-Seq peaks fell in introns (48.7% in K562 and 50.2% in HL60) and 50–500 kb from the TSS (Supplementary Fig. 5b). After assigning the peaks located within 5 kb of the TSS to their corresponding genes, we found that HLX was bound to

2135 and 6838 genes in K562 and HL60 cells, respectively. 1689 genes were found in both cell lines (hg.t., $P < 0$) (Supplementary Fig. 5c and Supplementary Data 4). Notably, 421 (19%, hg.t., $P < 7.044E-10$) and 1431 (21%, hg.t., $P < 1.088E-53$) bound genes in K562 and HL60 cells, respectively, also showed differential ATAC-Seq peaks in zebrafish. GO, IPA, Cytoscape, and GREAT analyses revealed that, similar to zebrafish, HLX directly regulates basic cellular processes, including metabolic pathways (Supplementary Fig. 5d and Supplementary Data 4). IPA analyses in K562 and HL60 cells demonstrated that HLX regulates mitochondrial and PPAR/RXR pathways (Fig. 4a). Indeed, multiple ETC genes, but also PPARδ, were directly bound by HLX in either or both cell types on regions with characteristics of enhancers as indicated by H3K4me1 and other histone marks (Fig. 4b and Supplementary Data 4). We confirmed these results by ChIP-qPCR on independent ChIP experiments with FLAG-tagged or HA-tagged constructs (Fig. 4c). We also performed ChIP-qPCR for HLX target genes on K562 cells carrying an endogenous 3xTy tag on the HLX gene (Fig. 4d). Deletion of one of the bound regions in the vicinity of the ATP11b gene using CRISPR-Cas9 technology confirmed that HLX-bound regions can affect gene expression (Supplementary Fig. 5e). Binding motifs for multiple transcription factors were identified in both K562 and HL60 cells (Fig. 4e and Supplementary Data 4). Importantly, independent motif analysis uncovered motifs for homeobox containing factors such as HMBOX1 (Fig. 4e and Supplementary Data 4). To unravel the chromatin landscape around the HLX-bound genomic regions, we analyzed available K562 chromatin data from the ENCODE database[38]. HLX-bound genomic regions were located mostly on open chromatin and active enhancers, as indicated by co-localization with active histone marks (Fig. 4f).

Next, we asked whether HLX binding is associated with changes in gene expression in human hematopoietic cells. To this end, we knocked out HLX in K562 cells using CRISPR-Cas9 technology and performed RNA-Seq analysis (Supplementary Fig. 5f). We found that 1324 genes were downregulated and 600 genes were upregulated in HLX knockout cells (>2-fold change, NBT, $P < 0.05$, Fig. 4g and Supplementary Data 5). Two hundred eighty-four (hg.t., $P < 9.874E-25$) and 731 (hg.t., $P < 4.460E-34$) deregulated genes were directly bound by HLX in K562 and HL60 cells, respectively (Supplementary Data 4). Expression of some ETC genes bound by HLX was increased, whereas that of PPARδ was decreased, consistent with our results in zebrafish (Fig. 4h). Additionally, we performed qPCR for ETC and PPARδ genes in K562 cells stably overexpressing an inducible HLX construct. These experiments showed that increased HLX expression leads to high PPARδ expression with concomitant lower expression of ETC genes (Supplementary Fig. 5g).

Here, we show that HLX directly regulates ETC and PPARδ gene transcription, and that this function is conserved from zebrafish to humans. These exciting results suggest that HLX controls myeloid differentiation, at least partly, through metabolic

**Fig. 2** hlx1 regulates the transcription of metabolic genes. **a** Heatmap of z-transformed normalized gene expression values from RNA-Seq performed on sorted endothelial/hematopoietic (fli:kaede+) cells from fli:hHLXOE embryos or endothelial cells (kdrl:GFP+) from hlx1MO at 48 hpf after unsupervised hierarchical clustering with Euclidean distance metric (see also Supplementary Data 1). **b** IPA analysis of deregulated genes in the RNA-Seq data from fli: hHLXOE or hlx1MO embryos at 48 hpf (NBT, $P \leq 0.05$ and $\geq 2$ fold change) (also see Supplementary Data 1). **c** Mean density plots of read distribution of lost and gained ATAC-Seq peaks between control and hlx1MO (see also Supplementary Data 2). **d** Digital genomic footprinting analysis showing average normalized Tn5 insertion profiles around footprinted motifs in merged ATAC peaks as indicated for control and hlx1MO (co-occurrence enrichment computation, $z = 3.583$ and $z = 13.241$, respectively). Insertions on the forward and reverse strands are indicated in red and blue, respectively. The numbers of motifs are indicated at the bottom of each panel. **e** Representative gene tracks from ATAC-Seq data of ETC and ppardb genes. **f** Heatmap of ETC gene expression in hlx1MO or fli:hHLXOE compared to control from the RNA-Seq analysis. All represented genes show differential chromatin accessibility in ATAC-Seq. **g–i** qPCR analysis of selected ppar genes in **g** endothelial cells from hlx1MO (data representative of two independant experiments, mean + s.d.) or (**h**) whole fli:hHLXOE embryos at 48 hpf ($n = 3$, mean + s.d., Student's t-test, *$P < 0.05$) or (**i**) whole Runx:hHLXOE embryos at 5 dpf ($n = 2$, mean + s.d.)

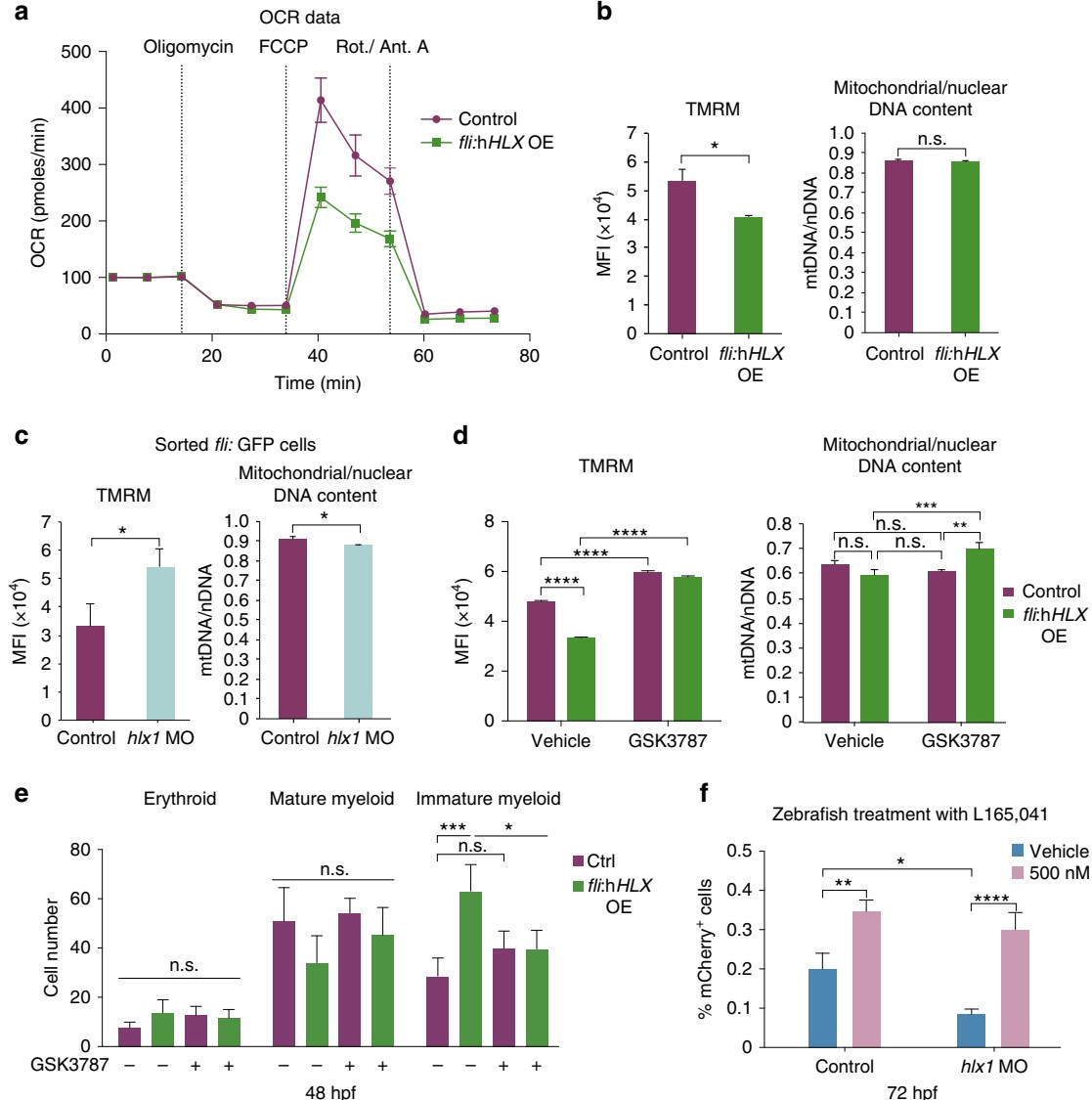

**Fig. 3** PPARδ modulation can rescue HSPC formation and myeloid differentiation in zebrafish. **a** Oxygen consumption rate (OCR) in control or *fli*:h*HLX*OE zebrafish. Representative plot from three independent experiments (mean ± s.d.). **b–c** Mitochondrial membrane potential measured by TMRM stain (left panel) and mitochondrial/nuclear DNA content analysis (right panel) in (**b**) control and *fli*:h*HLX*OE embryos or in (**c**) endothelial/hematopoietic cells (*fli*: GFP positive) of control and *hlx1*MO embryos at 48 hpf (*n* = 3, mean + s.d., Student's *t*-test, *$P < 0.05$). Graphs for TMRM depict median fluorescence intensity (MFI). The ratio of mitochondrial DNA (mtDNA) vs. nuclear DNA (nDNA) was measured at the same time (*n* = 3, mean + s.d., Student's *t*-test, *$P < 0.05$, n.s. non-significant). **d** Mitochondrial membrane potential at 48 hpf (left panel) in control or *fli*:h*HLX*OE embryos after GSK3787 treatment and corresponding ratio of mtDNA/nDNA (right panel) (*n* = 3, mean + s.d., ANOVA test, **$P < 0.01$, ***$P < 0.001$, ****$P < 0.0001$). **e** Cell count of zebrafish CHT smears from control or *fli*:h*HLX*OE embryos at 48 hpf after treatment with GSK3787 (*n* = 4, mean + s.d., Student's *t*-test, *$P < 0.05$, ***$P < 0.001$). **f** Percentage of *Runx:mCherry* HSPCs in *Tg(Runx:mCherry)* control or *hlx1*MO embryos at 72 hpf after L165,041 (PPARδ agonist) treatment (*n* = 3, mean + s.d., ANOVA test, *$P < 0.05$, **$P < 0.01$, ****$P < 0.0001$)

regulation. The combination of high *PPARδ* expression with low respiratory chain activity resembles the effects of pathways involved in controlling LT-HSC stemness[28,29,32] and could explain why *HLX*-overexpressing cells fail to terminally differentiate.

**HLX overexpression leads to elevated AMPK and autophagy.** HLX is particularly highly expressed in M4 and M5 AML leukemias[21]. We therefore selected THP1 cells deriving from a patient with AML (M5 subtype) to further explore the metabolic function of HLX. ChIP in *HLX*-overexpressing THP1 cells revealed that HLX binds to 5827 genes (Supplementary Data 4).

HLX was often found bound to intronic (42.24%) and intragenic (44.7%) regions, and specifically to ETC and PPARδ genes (Fig. 5a). HLX peaks were enriched for H3K27ac, a marker of active enhancers and promoters (Fig. 5b). Comparison of HLX ChIP in K562, HL60, and THP1 cells showed that 745 (12.8%, hg.t., $P < 7.838E-46$) and 1767 (30%, hg.t., $P < 6.676E-18$) of genes bound by HLX in THP1 cells were also bound in K562 and HL60 cells, respectively. To investigate whether the bound regions of HLX in THP1 cells represent open chromatin regions in HSCs and/or preleukemic and leukemic HSCs of AML patients, we used publicly available ATAC-Seq data[39] and selected randomly one donor for each condition. Strikingly, all HLX-bound regions fall into accessible chromatin regions in HSCs, pre-leukemic, and

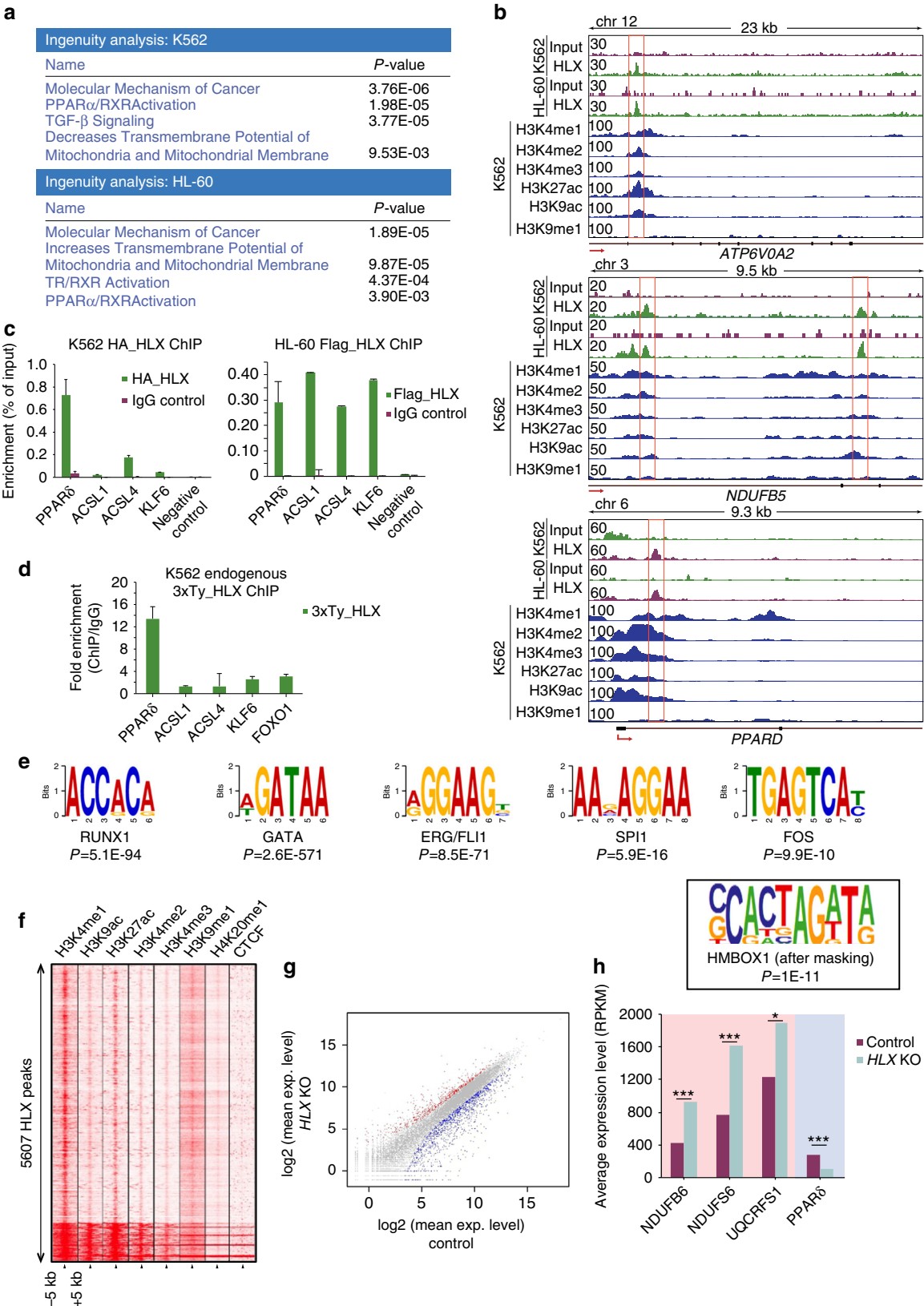

**e**

RUNX1
*P*=5.1E-94

GATA
*P*=2.6E-571

ERG/FLI1
*P*=8.5E-71

SPI1
*P*=5.9E-16

FOS
*P*=9.9E-10

HMBOX1 (after masking)
*P*=1E-11

leukemic HSCs (Fig. 5c). Thus, it is possible that perturbation of HLX binding affects genes implicated in HSCs or leukemic transformation.

Consistent with our previous results, ETC genes were down-regulated and PPARδ was upregulated at the protein level in THP1 cells (Fig. 5d). Moreover, maximal respiratory capacity was reduced, and lower levels of reactive oxygen species (ROS) were produced, in HLX-overexpressing THP1 cells (Fig. 5e, f). As PPARδ is a well-established regulator of fatty acid metabolism, we performed carbon tracing in control and HLX-overexpressing

THP1 cells cultured with $^{13}$C-glucose. Indeed, we found increased incorporation of glucose-derived carbon in citric acid that can be used to produce fatty acids, which was also reflected in palmitic acid, fatty acid C18, and stearic acid (Supplementary Fig. 6).

To uncover the mechanisms downstream of ETC gene downregulation by HLX, we took a candidate approach and examined the AMPK pathway. It has recently been proposed that mitochondrial dysfunction can lead to AMPK activation[40]. Moreover, AMPK is a well-established sensor of metabolic stress and its activation results in elevated autophagy[41,42]. We found that AMPKα and phospho-AMPKα (p-AMPKα) are upregulated in HLX-overexpressing THP1 cells (Fig. 5d). Additionally, the protein levels of LC3-II, an autophagosome marker, are markedly increased in these cells upon chloroquine treatment (Fig. 5g).

Together, these results suggest that *HLX* overexpression in AML cells affects mitochondrial metabolism and fatty acid synthesis possibly through the upregulation of *PPARδ* gene expression. Additionally, *HLX* overexpression, possibly through downregulation of ETC genes, results in AMPK activation and autophagy.

**HLX regulates the metabolic state of CD34$^+$ human cells**. Our findings could have important implications for patients with hematopoietic disorders, including AML. We therefore performed colony-forming unit (CFU) assays on human CD34$^+$ HSPCs to measure the effects of HLX modulation in normal hematopoiesis. Consistent with our zebrafish results and with published data on mouse HSPCs[21], *HLX* knockdown (sh-*HLX*) caused a significant reduction in the number of hematopoietic colonies, whereas HLX overexpression (CD34$^+$ *HLX*) caused the opposite phenotype and large myeloid colonies (Fig. 6a, b and Supplementary Fig. 7a). Culturing CD34$^+$ *HLX* in myeloid differentiation media resulted in a maturation block, as revealed by the accumulation of early granulocyte-monocyte progenitors (early GMPs) and the relatively reduced numbers of mature monocytes and granulocytes (Fig. 6c and Supplementary Fig. 7b). To assess whether *PPARδ* and ETC genes are regulated by HLX in CD34$^+$ cells, we performed RNA-Seq experiments upon *HLX* knockdown or overexpression. Due to the high variability in primary cells we considered all the genes that have at least 1.5-fold change independently of *P*-value. The expression of many ETC genes was clearly upregulated upon sh-*HLX* and downregulated in CD34$^+$ *HLX* cells (Fig. 6d and Supplementary Data 5). Selected targets were validated with qPCR (Supplementary Fig. 7c). Thus, the function of HLX in hematopoiesis and its target genes are conserved in human primary hematopoietic cells.

To further determine whether the metabolic function of HLX is conserved in CD34$^+$ cells, we measured OXPHOS. Similar to zebrafish, increased levels of *HLX* in human CD34$^+$ cells led to a reduction in OCR, particularly spare respiratory capacity (Fig. 6e). Moreover, although *HLX* overexpression did not cause significant changes on the OXPHOS to extracellular acidification rate (ECAR, representative of glycolytic rate) ratio, *HLX*-overexpressing cells tended to have a lower ratio (Supplementary Fig. 7d),

suggesting a metabolic adaptation by engagement of glycolysis. TMRM staining shows that CD34$^+$ *HLX* cells have decreased mitochondrial membrane potential, independently of mitochondrial mass (Fig. 6f). However, upon differentiation toward myeloid cells, CD34$^+$ *HLX* cells exhibited significantly higher OXPHOS than control cells and a tendency to higher OCR/ECAR ratio (Supplementary Fig. 7e, f).

Our results in THP1 cells showed that *HLX* overexpression is followed by AMPK activation. To determine whether AMPK activation affects myeloid maturation, we induced differentiation of human CD34$^+$ HSPCs in the presence or absence of metformin, an AMPK activator that also blocks mitochondrial complex I thus mimicking the effect of HLX[43]. Notably, metformin induced a myeloid differentiation block in CD34$^+$ cells (Fig. 6g). These results suggest that metabolic manipulation can indeed be the underlying reason for the hematopoietic phenotypes caused by HLX deregulation.

We next assessed whether pharmacological modulation of PPARδ activity rescues the myeloid differentiation phenotypes caused by HLX. Indeed, treatment with a PPARδ antagonist relieved the myeloid differentiation block in CD34$^+$ *HLX* cells (Fig. 7a). Moreover, the inability of sh-*HLX* cells to form colonies in CFU assays was partially rescued by a PPARδ agonist (L165,041) (Fig. 7b). This agonist also rescued the increased mitochondrial membrane potential observed in sh-*HLX* cells (Fig. 7c). To investigate this further, we identified genes affected by HLX and potentially regulated by PPARδ using publicly available PPARδ-binding data in human macrophages[44]. We compared these data to our RNA-Seq from human CD34$^+$ cells and found that 399 (hg.t., $P < 0.006$) deregulated genes upon *HLX* knockdown that can potentially be bound directly by PPARδ (Supplementary Data 6). IPA analysis on these genes showed involvement in FAO I (rtFET, $P < 2.26$ E-04) and AMPK signaling (rtFET, $P < 5.24$ E-03) (Supplementary Data 6).

**AMPK inhibition causes reduced viability of AML cell lines**. To investigate the potential role of PPARδ and AMPK in promoting AML downstream of HLX, first, we analyzed the expression levels of *HLX* and *PPARδ* by qPCR in various AML cell lines and one CML cell line, K562. As expected, THP1, a M5 subtype leukemia, exhibited the highest levels of *HLX* expression[21] but also PPARδ expression (Fig. 8a). PPARδ protein was only detectable in THP1 cells (Fig. 8b). AMPKα and pAMPKα expression was detected in all cell lines without any noticeable differences (Fig. 8b). TMRM and autophagy were variable between cell lines (Fig. 8c, d). Next, we asked whether PPARδ and AMPK inhibitors could push the AML lines and/or K562 cells toward myeloid maturation or affect their viability. PPARδ antagonists (GSK3787) had no significant impact on either the viability or the differentiation of AML cell lines and K562 cells (Fig. 8e). However, AMPK inhibition with dorsomorphin significantly reduced the viability of all but one (HL60) AML cell lines tested and K562 cells (Fig. 8f). It is important to note that dorsomorphin had only a mild effect on the viability of CD34$^+$ myeloid progenitor cell populations (Fig. 8f).

**Fig. 4** Direct binding of HLX on metabolic genes. **a** IPA analysis of K562 and HL60 HLX ChIP-Seq peaks (*P*-values rtFET, see also Supplementary Data 4). **b** Representative gene tracks of HLX-bound genomic regions in K562 and HL60 cells, together with publicly available data for histone modifications in K562 cells. **c** ChIP-qPCR validation of selected HLX peaks in K562 and HL60 cells ($n = 2$, mean + s.d.). **d** ChIP-qPCR validation upon ChIP of the endogenous 3xTy-tagged HLX in K562 cells ($n = 2$, mean + s.d.). **e** Representative motifs identified in HLX-bound genomic regions in K562 cells (*P*-values binomial test, see also Supplementary Data 4). HOX motif is shown in the frame. **f** Heatmap of ChIP-Seq signals, comparing HLX-bound regions in K562 cells to publicly available CHIP-Seq data for histone marks in K562 cells from the ENCODE database. CTCF was used as negative control. **g** Gene expression in control samples plotted vs. gene expression in *HLX* KO K562 cells. Differentially expressed genes (>2-fold, NBT, $P < 0.05$) are colored in red and blue for upregulated and downregulated genes, respectively (also see Supplementary Data 5). **h** Average expression levels of selected genes from RNA-Seq of K562 *HLX* KO cell line (Student's *t*-test, *$P < 0.05$, ***$P < 0.001$).

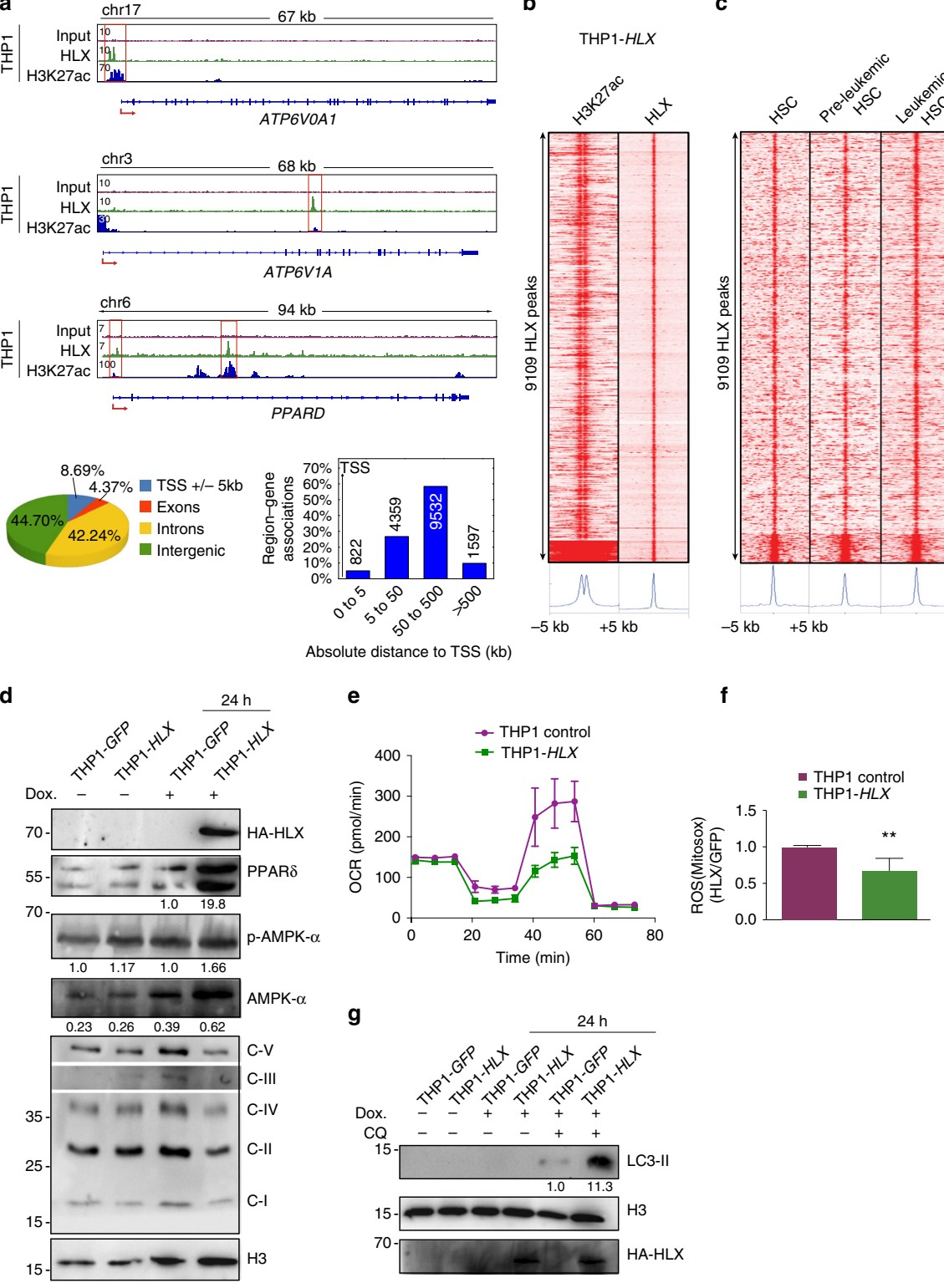

Thus, we show here that AMPK inhibition decreases the viability of AML cell lines. Our study depicts HLX as a novel metabolic regulator in both normal and malignant cells (Fig. 8g).

## Discussion

Recent evidence showing that the H2.0-like HLX is implicated in many malignancies highlights the importance of understanding the function and identifying the targets of this transcription factor[21,45,46]. In agreement with previous reports[21], in the present study we demonstrate in zebrafish models that HLX affects myeloid differentiation, and we recapitulate these results in human HSPCs for the first time. We provide evidence that this regulation occurs, at least partly, through direct modulation of metabolic pathways by HLX. Specifically, we show that HLX directly regulates several metabolic genes, including *PPARδ* and genes of the mitochondrial ETC. In agreement with our results, it

was recently shown by Huang et al. that HLX controls a systematic switch from white to brown fat through metabolic gene regulation, including PPARs and genes that control mitochondrial biogenesis[24]. However, in that study HLX was found to positively regulate both mitochondrial biogenesis and PPARs, in contrast to our study where ETC genes are downregulated upon HLX overexpression. It will be interesting to study whether diverse HLX-interacting partners could account for these differences.

Recently, metabolism has emerged as a critical regulator of HSCs. LT-HSCs are quiescent and rely mostly on anaerobic glycolysis rather than OXPHOS[47,48]. A number of studies have shown that low mitochondrial activity is necessary to maintain the quiescent state and the self-renewal capacity of LT-HSCs and protect them from oxidative stress[28,49–53]. The importance of reduced mitochondrial activity for HSC maintenance has also been demonstrated in human CD34[+] HSPCs[54–56]. Antagonism of PPARγ signaling enhances glycolysis and leads to expansion of human HSPCs[57]. Moreover, recent evidence revealed that PPARδ, a regulator of fatty acid metabolism, plays an essential role in maintaining HSC stemness by regulating mitophagy and promoting HSC asymmetric cell divisions[32,58]. During differentiation, PPARδ signaling is downregulated leading to a gradual increase in mitochondrial mass and symmetric commitment of HSC daughter cells[32]. Thus, PPARδ, but also HLX that controls its expression, may constitute a metabolic switch for regulating HSC cell fate. It is interesting to speculate, based on our results, that HLX is a gatekeeper of HSC identity by maintaining their glycolytic state. We also show that HLX overexpression leads to low spare respiratory capacity, a characteristic of AML cells[59]. Both these results could be used in the future to better understand the implication of HLX in AML. However, since both HLX and PPARδ exert many functions besides metabolic regulation, we need to fully understand the precise mechanisms underlying our rescue experiments by PPARδ modulation. Interestingly, since PPARδ, like HLX, is overexpressed in a subset of M5 type-monoblastic AML cases[60], it is conceivable that PPARδ inhibition could play a role in AML. Our results do not support this hypothesis, but further investigations should shed more light on a potential role of PPARδ in AML.

Also of note, we show that metformin, which blocks mitochondrial complex I and activates AMPK[43], thereby mimicking the metabolic function of HLX, can affect myeloid differentiation. This finding proves that metabolic regulation can indeed be the direct mediator of HLX functions and, at least in part, causative for the phenotype. Metformin has been used as an anticancer therapy in many malignancies, including AML[61]. Based on our findings, it is pertinent to fully understand the role of metformin in physiological and pathological conditions.

Finally, we found that HLX overexpression leads to activation of AMPK. AMPK does not seem to play a role in HSCs during homeostasis, transplantation, or under metabolic stress[62], but protects leukemia initiating cells from metabolic stress[63]. This suggests AMPK as an ideal potential target for the treatment of leukemia without affecting normal cells. Indeed, we found that AMPK inhibition has a strong effect on the survival of AML cell lines. AMPK inhibition is also successful in eliminating MLL-rearranged B-cell acute lymphoblastic leukemia[64]. However, other studies suggest that AMPK acts synergistically with mTORC1 and causes lethality in AML cells[65]. Further research with samples from human patients and specific mouse models are needed to clarify these discrepancies.

Our study points to differential requirements and regulatory mechanisms between normal and leukemic cells by the same transcription factor and identifies HLX as a new player in metabolic regulation in hematopoiesis.

## Methods

**Zebrafish maintenance.** The zebrafish (*D. rerio*) strain Tübingen (*Tü*) and all zebrafish transgenic lines used in this study were maintained in the animal facility of the Max Planck Institute of Immunobiology and Epigenetics. The sample size for the animal experiments was chosen according to the following paper[66]. No animals were excluded from this study and no randomization was used. Only 1–5 dpf embryos were used in this study and sex was not determined at these stages. All animal experiments were performed in accordance with relevant guidelines and regulations, approved by the review committee of the Max Planck Institute of Immunobiology and Epigenetics and the Regierungspräsidium Freiburg, Germany (license Az 35-9185.81/G-14/95).

**Zebrafish morpholino injections and rescue experiments.** Embryos were injected (PV820 Pneumatic PicoPump, World Precision Instruments) at the one-cell stage with 8 ng of standard morpholino or 8 ng of *hlx1* translational or 12 ng splicing anti-sense morpholino that have been previously described[26] (Gene Tools, Philomath, OR). Stock solutions were diluted as recommended by the manufacturer. The sequence for the *hlx1* translational anti-sense morpholino: 5′-AGCCGAACAATACGGCAGTCCACAGG-3′; splicing anti-sense morpholino: 5′-GATTAAATTAGCGTCTTACCTCTCA-3′; standard oligo: 5′-CCTCTTACCTCAGTTACAATTTATA-3′.

For the rescue experiments one-cell stage *Tg(Runx:mCherry)*[27] embryos were injected with 12.5 or 25 pg of *fli:HLX* or *Runx:HLX* constructs followed by injection of *hlx1* morpholino as described above. Injected embryos were grown until 72 hpf, manually dechorionated and embedded in 1% low melting agarose, containing 0.04 mg/mL tricaine. Caudal hematopoietic tissue was imaged using Zeiss Apotome2 microscope at 10× magnification. Flow cytometry of mCherry-positive cells is described in the section "Preparation of zebrafish cells, flow cytometry, and cell sorting". WISH staining and analysis, constructs and generation of transgenic zebrafish lines are described in Supplementary Material and Methods.

**Preparation of zebrafish cells, flow cytometry, and cell sorting.** Embryos were incubated in 0.5 mg/mL Liberase TM (Roche) solution for 30 min at 37 °C, then dissociated and resuspended in PBS-5% fetal bovine serum (FBS), and used for cell sorting, flow cytometry, seahorse assay, qPCR, and RNA-Seq experiments. Cell sorting was performed using Influx (BD Biosciences). For all experiments cell-sorting purity was over 85%.

**May–Grünwald–Giemsa staining of zebrafish blood.** For CHT smears, 48 hpf (*fli*:hHLXOE) or 5 dpf (*Runx*:hHLXOE) embryos were placed in 0.9% NaCl with 0.04 mg/mL tricaine and the tails were isolated at the level of the cloaca/end of the yolk sac extension and incubated with Liberase TM (at 1:65 dilution in 0.9% NaCl, Roche) with 0.04 mg/mL tricaine for 20 min at 37 °C. FBS was then added to a final concentration of 10%, to stop enzymatic digestion. The tails were triturated and then passed through a 40 μM mesh filter. The smears of dissociated cells were prepared by cytospin followed by May–Grünwald–Giemsa staining. Cells were imaged using Zeiss Axio Imager microscope with 100× objective.

**Fig. 5** AMPK and autophagy activation in HLX-overexpressing cells. **a** Representative gene tracks of HLX and H3K27ac-bound genomic regions in THP1 cells. Location annotation of HLX-bound regions across the genome. **b** Heatmap of ChIP-Seq signals comparing HLX-bound regions to H3K27ac regions in THP1 cells. **c** Heatmap comparing HLX-bound regions in THP1 cells to ATAC-Seq regions of HSCs, pre-leukemic HSCs, and leukemic HSCs. **d** Western blot analysis of HA-HLX, PPARδ, AMPKα, and p-AMPKα, mitochondrial ETC complexes and Histone 3 (H3) as loading control. Cells were untreated or treated with doxycycline for 24 h to induce *HLX* expression. Representative immunoblots of at least three independent experiments. Quantifications were performed by FIJI software software and are shown below the blots as a ratio to H3. The five OXPHOS complexes are depicted as C-I to V. **e** Oxygen consumption rate (OCR) in control or *HLX*-overexpressing THP1 cells. Representative plot from three independent experiments (mean ± s.d.). **f** Bar graph depicting mitochondrial ROS production in control and THP1-*HLX* cells ($n = 3$, mean + s.d., Student's *t*-test, **$P < 0.01$). **g** Western blot analysis of LC3-II, HA, and H3 in control and THP1-*HLX* cells non-induced, induced with Doxycycline (Dox) with or without the addition of Chloroquine (CQ). Representative immunoblots of three independent experiments. Quantification was performed by Fiji software and is shown below the blot as a ratio to H3

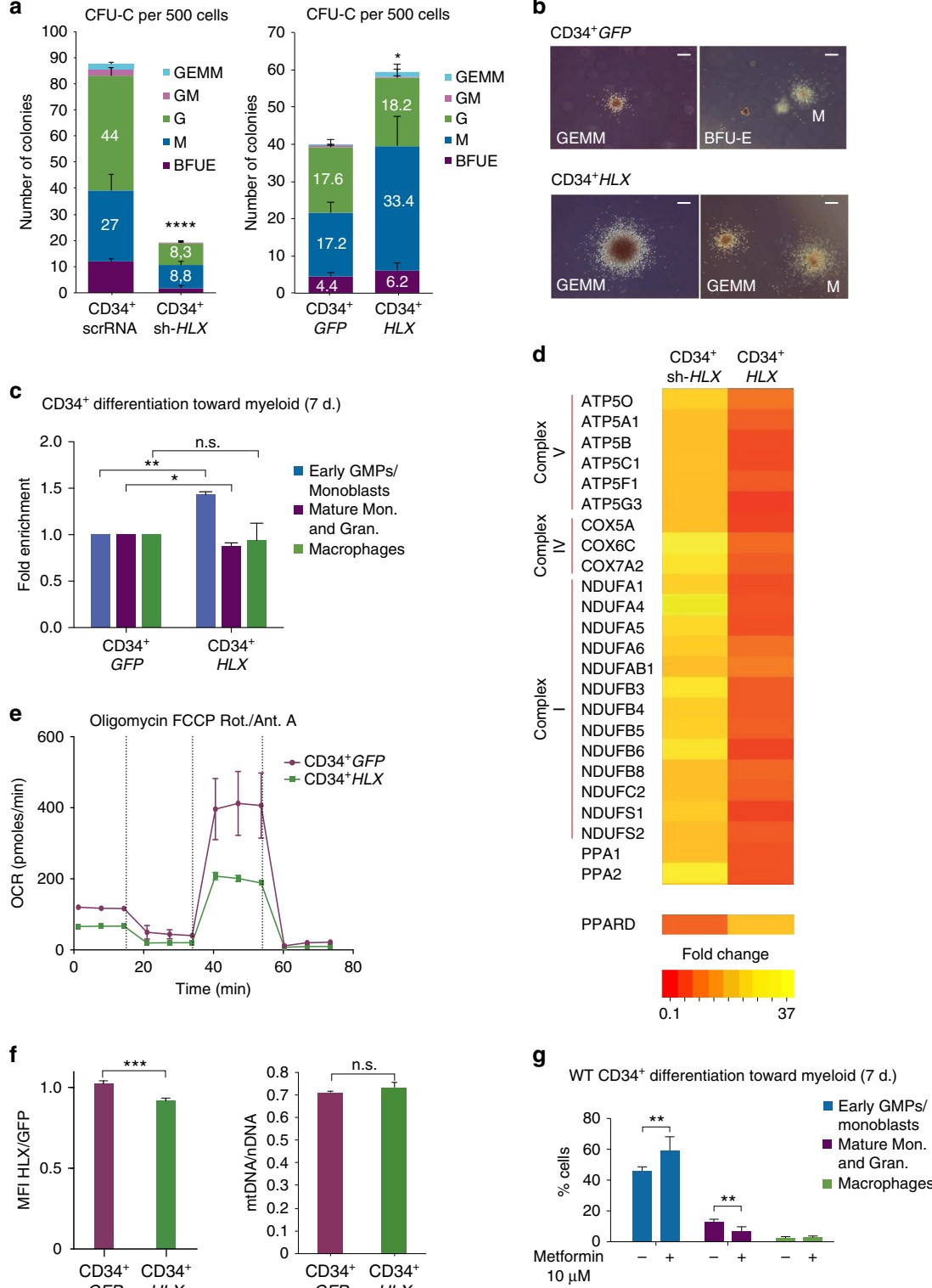

**TUNEL assay**. Whole-mount TUNEL staining of developmentally staged control, morphant, and overexpression embryos was performed using the in situ cell death detection kit with fluorescein (Roche Applied Science, 11684795910). Embryos were then embedded in 1% low melting agarose and imaged with a Zeiss LSM780 confocal microscope and a 10× objective.

**EdU labeling**. Cell proliferation in zebrafish was assayed using the Click-iT EdU Alexa Fluor 647 Imaging Kit (Thermo Fisher Scientific). Briefly, zebrafish embryos at 48 hpf were dechorionated and incubated with Liberase TM as described in the section "Preparation of zebrafish cells, flow cytometry, and cell sorting".

Dissociated cells were spun down, resuspended in EdU staining solution, and incubated for 30 min at 37 °C. Cells were fixed and permeabilized, followed by EdU detection reaction, according to the manufacturers's instructions. Cells were then analyzed by flow cytometry (BD LSRFortessa).

**Cell line and primary cell maintenance**. K562 and SKNO1 cells were maintained in Iscove's Modified Dulbecco's Medium (IMDM, Gibco) supplemented with 10% FBS (Sigma, F7524) and 1% Penicillin Streptomycin (P/S, 100 U/mL penicillin with 100 μg/mL streptomycin, Gibco). KG1 cells were maintained in IMDM 20% FBS, 1% P/S. HL60, NB4, and THP-1 cells were maintained in RPMI 1640 medium

**Fig. 6** Metabolic role of HLX in human CD34+ cells. **a** Average number of colonies in CFU-C assays performed in CD34+ cells infected with control scrRNA or sh-*HLX* (left panel) or in CD34+ cells overexpressing *GFP* (CD34+ *GFP*) or *HLX* (CD34+ *HLX*) (right panel) (n = 3, three technical replicates each time, mean + s.d., Student's *t*-test, *P < 0.05, ****P < 0.0001). GEMM Granulocyte, Erythrocyte, Monocyte, Megakaryocyte, GM Granulocyte, Macrophage, G Granulocyte, M Macrophage, BFUE Burst-forming unit-erythroid. **b** Representative images of GEMM, BFUE, and M colonies from CFU-C assays in CD34+ *GFP* or CD34+ *HLX* cells. Scale bar: 50 μM. **c** Flow cytometric analysis representing elevated numbers of immature myeloid cells in CD34+ *HLX* cells differentiated toward the myeloid lineage for 7 days (n = 3 + s.d., Student's *t*-test, *P < 0.05, **P < 0.01). **d** Heatmap representing the fold change of selected genes from RNA-Seq of CD34+ sh-*HLX* and CD34+ *HLX* cells compared to their respective controls. **e** Oxygen consumption rate in CD34+ *GFP* or *HLX* cells before differentiation (n = 3, mean + s.d.). **f** Mitochondrial membrane potential measured by TMRM in CD34+ *GFP* or CD34+ *HLX* cells (left panel) (n = 6, mean + s.d., Student's *t*-test, ***P < 0.001) and the corresponding mitochondrial vs. nuclear DNA ratio (right panel) (n = 3, mean + s.d., Student's *t*-test n.s.). **g** Flow cytometric analysis representing numbers of immature myeloid cells upon addition of metformin to wild-type CD34+ cells cultured in myeloid differentiation media for 7 days (n = 3, mean + s.d., Student's *t*-test, **P < 0.01)

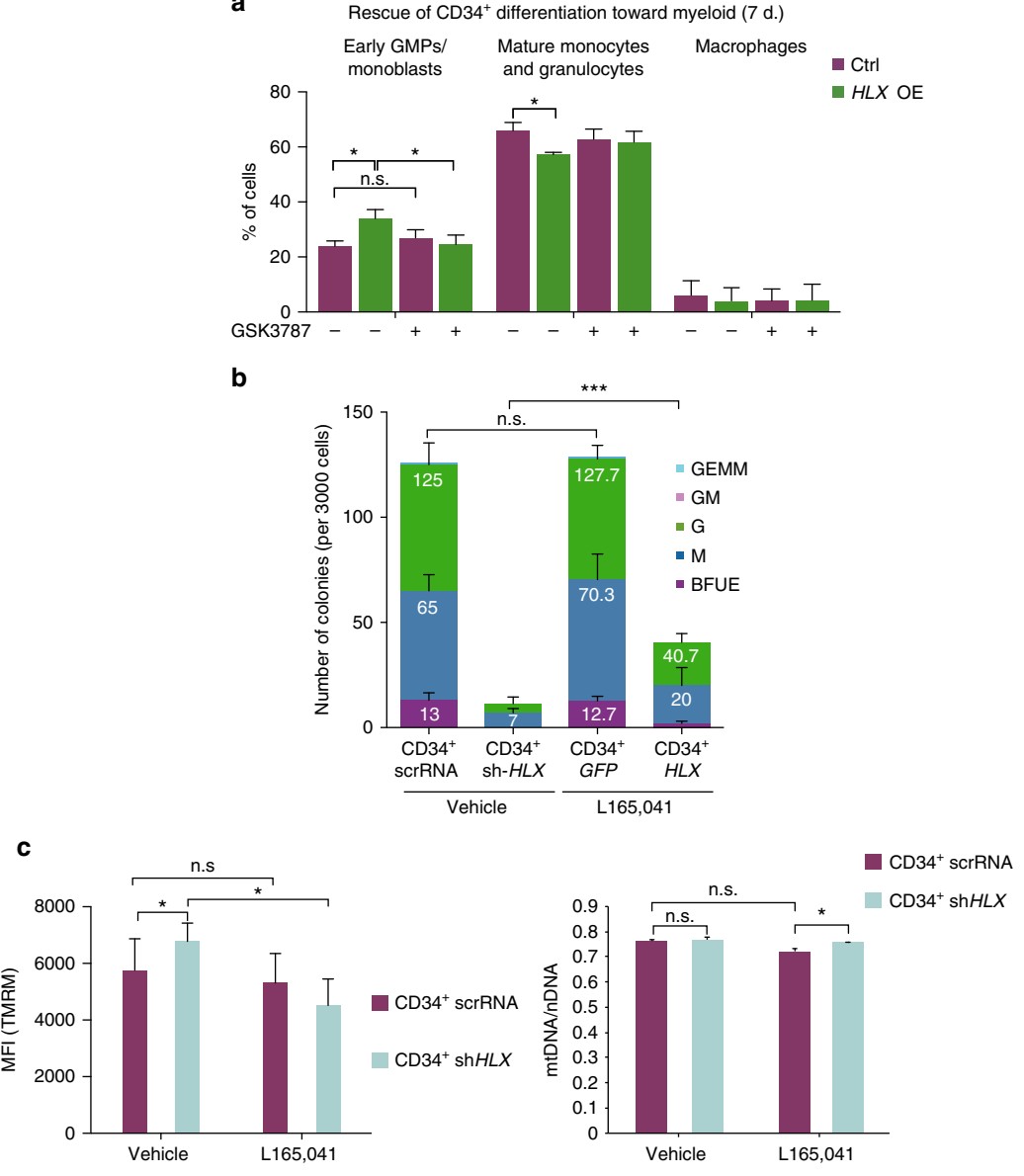

**Fig. 7** PPARδ modulation rescues hematopoietic phenotypes in human primary cells. **a** Flow cytometric analysis of CD34+ *GFP* or CD34+ *HLX* cells cultured in differentiation media for 7 days in the presence or absence of the PPARδ antagonist GSK3787 depicts rescue of the myeloid differentiation block after treatment (n = 3, Student's *t*-test, *P < 0.05, mean + s.d.). **b** Average number of colonies in CFU-C assays performed in CD34+ cells infected with control scrRNA or sh-*HLX*, treated or not with L165,041 (n = 2, three technical replicates each time, mean + s.d., Student's *t*-test, ***P < 0.001). **c** Mitochondrial membrane potential measured by TMRM (left panel) and respective mitochondria vs. nuclear DNA ratio (right panel), in CD34+ infected with scrRNA or sh-*HLX* treated or not with L165,041 (n = 3, mean + s.d., Student's *t*-test, *P < 0.05)

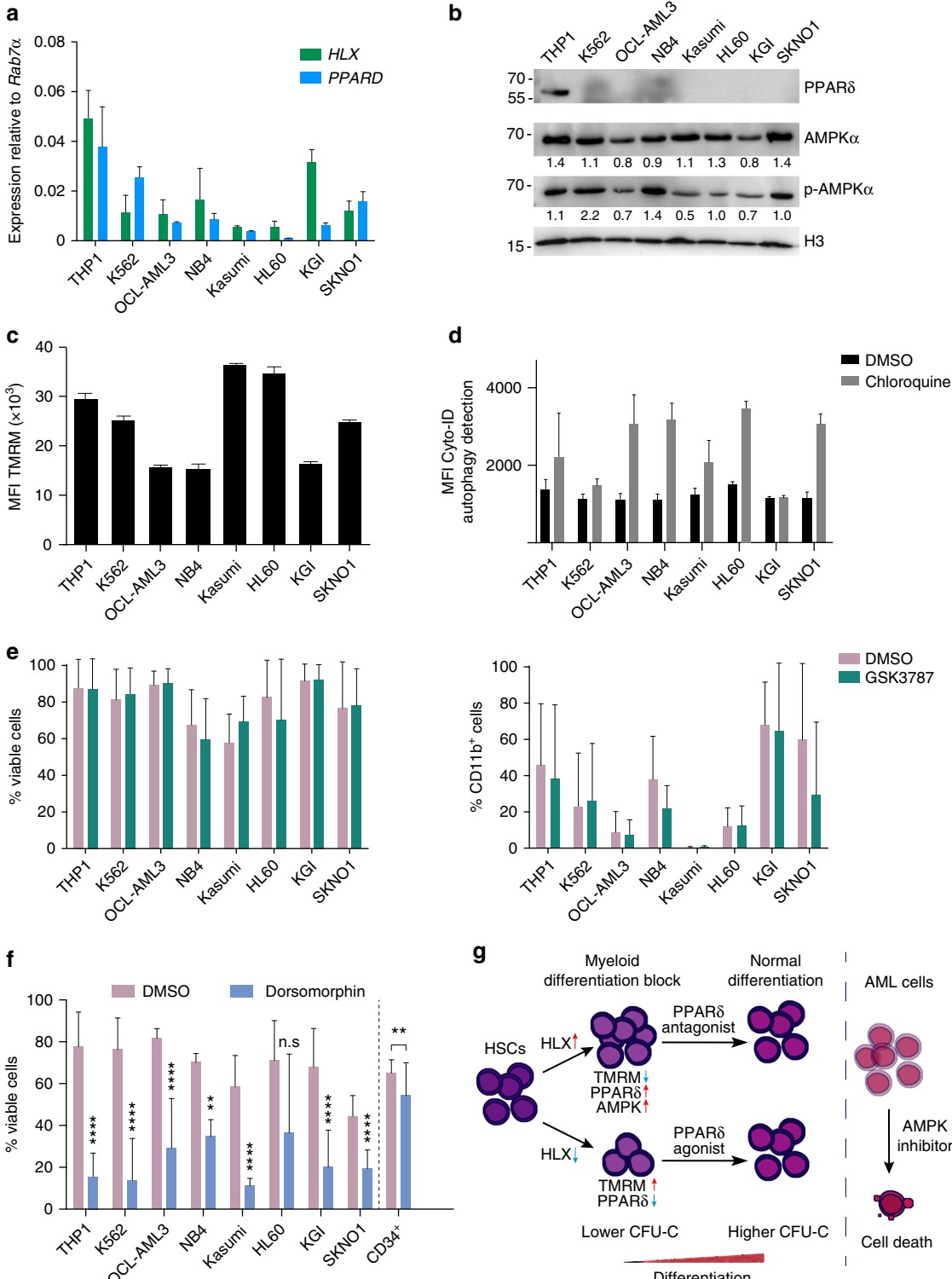

**Fig. 8** AMPK inhibition causes lethality in AML cell lines. **a** qPCR expression analysis of *HLX* and *PPARδ* in AML cell lines and K562 cells ($n = 2$, mean + s.e.m.). **b** Representative western blot analysis for PPARδ ($n = 4$), AMPKα ($n = 3$), p-AMPKα ($n = 2$) and H3 as loading control in different AML cell lines and K562 cells. Quantifications shown below the panels, as a ratio to H3, were performed by FIJI software. **c** Mitochondrial membrane potential measured by TMRM in AML cell lines and K562 cells. MFI median fluorescence intensity (representative of two independent experiments, mean + s.d.). **d** Flow cytometry analysis of autophagic flux in different AML cell lines and K562 cells incubated with DMSO or chloroquine. MFI median fluorescence intensity (representative of two independent experiments, mean + s.d.). **e** Measurement of viable cells (left panel) or CD11b staining as a marker of myeloid differentiation (right panel) with or without the PPARδ antagonist GSK3787 ($n = 5$, mean + s.d.). **f** Measurement of viable cells after incubation of different AML cell lines and K562 cells or WT CD34+ cells with or without dorsomorphin ($n = 3$, mean + s.d., Student's *t*-test, **$P < 0.01$, ****$P < 0.0001$). **g** Model of HLX function in hematopoiesis

(Gibco) supplemented with 10% FBS, 1% P/S. Kasumi cells in RPMI 1640 medium with 20% FBS, 1% P/S. The cell line OCI-AML3 was maintained in Alpha MEM (Gibco) supplemented with 20% FBS, 1% P/S. All media were supplemented with 2 μM glutamine. K562 cells were verified from Eurofins and from RNA-Seq experiments and THP1 cells verified by next generation sequencing experiments. Cell lines were mycoplasma free as tested by qPCR. Cell lines were purchased from Sigma or ATCC or Lonza. Cell line generation and constructs can be found in the Supplementary Information.

Human CD34[+] cells, isolated from the peripheral blood of granulocyte colony-stimulating factor mobilized healthy volunteers, were purchased from the Fred Hutchinson Cancer Research Center. The cells were maintained as previously described[67]. Briefly cells were cultured in StemSpan SFEM (StemCell Technologies), supplemented with 2% P/S and cytokine mix: m-SCF 100 ng/mL, hFLT3 100 ng/mL, hIL-3 20 ng/mL, and hIL-6 20 ng/mL (expansion medium). Cells were kept at $1 \times 10^5$–$1 \times 10^6$ cells/mL densities.

**Lentiviral particle production**. 293T cells were transfected with lentiviral plasmids (packaging vectors with *GFP*-HA or *HLX*-HA or sh-*HLX* or scrRNA) mix using the polyethylenimine (PEI) method. Briefly, 2 h before the transfection, fresh 2% FBS containing medium was added. For 9 cm culture dishes, 5 μg of total DNA mix (construct, pPAX2, and pMDG.2 were used at a ratio of 10:7.5:3, respectively) was diluted with plain DMEM (Gibco) up to 520 μL. 30 μL of 1 mg/mL PEI reagent was then added and the mix was vortexed shortly twice. The mix was incubated for 10 min at RT before adding drop wise to 293T cells. The medium was changed after 8 h. Supernatants containing lentiviral particles were collected at 48, 72, and 96 h post transfection and concentrated using Lenti-X concentrator (TAKARA). All lentiviral experiments were performed in S2 laboratory with permission from German authorities: Regierungspräsidium Tübingen (57-3/8817.40-020).

**Electroporation or infection of CD34[+] cells**. *HLX* overexpression: 1 day after thawing, 1 million cells were electroporated using the Human CD34[+] Cell Nucleofector kit (Lonza) with 5 μg pCMV6-AC-*HLX* (Origene, NM_021958) or as a control 5 μg of pEGFP-C1 plasmids. A day after electroporation CD34[+] *GFP* and *HLX* cells were selected with neomycin (0.8 mg/mL) and expanded for 4 more days.

*HLX* knockdown: To generate shRNA against *HLX* (sh-*HLX*), we cloned annealed oligos (Supplementary Table 1) into the pLKO.1-TCR vector (a gift from David Root; Addgene plasmid #10878) digested with AgeI and EcoRI. As a control, a non-hairpin pLKO.1-TCR control (scrRNA) (a gift from David Root; Addgene plasmid #10879)[68] was used. For the generation of scrRNA and sh*HLX*, human CD34[+] cells were infected after 1-day expansion in expansion medium (see section "Cell line and primary cell maintenance"). Cells were placed on retronectin-coated (TAKARA) plates and transduced with concentrated virus at a multiplicity of infection of 5 in expansion medium. Cells ($1 \times 10^6$ cells/mL; 1 mL per well in 6-well plates) were transduced three times using spinoculation (6 μg/mL polybrene, $800 \times g$, 90 min) at 6–12 h intervals for 2 consecutive days. Then cells were washed five times in PBS and transefered in expansion medium containing puromycin (1 μg/mL) for 3 more days.

**CFU-C assays**. The CFU-C assays were performed by plating 500 or 3000 CD34[+] cells per plate in CFU-C media (R&D Systems, HSC003), according to the manufacturer's instructions. Colonies consisting of at least 40 cells were counted after 15 days at 37 °C and 5% $CO_2$. CFU-C colonies were counted blindly regarding control and experimental samples.

**Myeloid differentiation**. Myeloid lineage-specific CD34[+] differentiation was carried out by culturing the cells in expansion media supplemented with 20 ng/mL GM-CSF (Peprotech, ref. 300-03). Cells were cultured at $1 \times 10^5$–$1 \times 10^6$ cells/mL densities and analyzed on day 7 and day 14, using CD34[+]-PerCP-Cy5.5 (1:60, clone 561), CD11b-PE-Cy7 (1:60, clone ICRF44), CD14-AlexaFluor700 (1:200, clone 63D3), CD16-APC (1:40, clone B73.1), and CD33-PE conjugated antibodies (1:60, clone P67.6) (all from BioLegend). Myeloid differentiation of AML cell lines was assessed using CD11b-PE-Cy7 (1:60, Biolegend clone ICRF44).

**Cell viability**. Viability was assessed by flow cytometry after staining with Hoechst 33258 (Life Technologies, 1 μg/ml; ref. H3569) in PBS supplemented with 2% FBS and 1 mM EDTA. After 20 min, cells were analyzed in a cell analyzer (BD, LSRFortessa). Viable cells were determined by Hoechst staining exclusion and quantified using FlowJo software (Tree Star, Inc.).

**Pharmacological treatments**. Zebrafish: For pharmacological treatment, embryos were manually dechorionated and incubated with the PPARδ agonist L165,041 (Cayman Chemical, ref. 9000249; 500 nM) or antagonist GSK3787 (BioVision, ref. 2400; 500 nM) from the 18 somite stage to 48 or 72 hpf.

Cells: For mitochondrial membrane potential rescue experiment CD34[+] scrRNA and sh-*HLX* cells were treated with vehicle (DMSO) or 100 nM PPARδ agonist L165,041, while CD34[+] *GFP* and *HLX* were treated with vehicle (DMSO) or 300 nM PPARδ antagonist GSK3787 for 21 h. For the CD34[+] differentiation

experiments, cells were treated with 300 nM antagonist GSK3787 (BioVision, ref. 2897-5), or 5 μM Dorsomorphin hydrochloride (Enzo, ref. ENZ-CHM141) for the last 48 h of the 7-day differentiation protocol, or 10 μM Metformin hydrochloride (Sigma, ref. PHR1084) for the last 4 days of the 7-day differentiation. For the HLX knockdown rescue experiments scrRNA and sh*HLX* cells were treated with 100 nM PPARδ agonist L165,041 for 7 days. Leukemic cell lines were treated for 48 h with 1 μM PPARδ antagonist GSK3787, or 5 μM Dorsomorphin.

**Detection of autophagic flux**. Autophagy was measured using the Cyto-ID® Autophagy detection kit (Enzo, ref. ENZ-51031), according to the manufacturer's instructions. Briefly, cells were grown at a density of $1 \times 10^6$ cells/mL for 16 h at 37 °C, in the presence of DMSO or Chloroquine (60 μM). Cells were washed with PBS, pelleted at 1000 rpm for 5 min, and incubated with cyto-ID green reagent for 30 min at 37 °C in the dark prior to flow cytometry analysis on LSRFortessa cell analyzer (BD Biosciences). Mean fluorescence intensity (MFI) was quantified using FlowJo software (Tree Star, Inc.).

**Real-time qPCR**. Total RNA was extracted from human or zebrafish cells using the RNA Clean & Concentrator-5 kit (Zymo Reasearch) or TRI Reagent, according to the manufacturer's instructions. cDNA was prepared with the SuperScript™ First-Strand Synthesis System for RT-PCR kit (Thermo Fisher Scientific). qPCR reactions were executed using Fast SYBR green Master Mix (Thermo Fisher Scientific) in a StepOnePlus Real-Time PCR machine (Applied Biosystem). Expression was plotted relative to *RAB7* (for human cells) and *cyyr1* (for zebrafish cells). All qPCR graphs in this study are representative of at least two independent experiments. qPCR primers can be found in Supplementary Table 2.

**Metabolic assays**. OCR and ECAR were measured using a 96-well XF or XFe extracellular flux analyzer (Seahorse Bioscience) in XF media (non-buffered RPMI medium 1640 containing 25 mM glucose, 2 mM glutamine, and 1 mM sodium pyruvate) containing the cytokine mix of expansion medium for CD34[+] experiments or 10% FBS for zebrafish experiments. The measurements were performed under basal conditions and in response to 1 μM oligomycin, 1.5 μM FCCP, and 100 nM rotenone combined with 1 μM antimycin A.

**Glucose tracing**. For metabolic tracing THP1 cells were cultured in glucose-free RPMI media with dialyzed serum supplemented with 11 mM D-[U13C] glucose and 2% P/S for 24 h. Cells were washed with ice cold 0.9% w/v NaCl buffer and metabolites were extracted twice with hot ethanol (70%) and analyzed by GC mass spectrometry (GCMS). Fractional contribution from exogenous substrates was calculated as described previously[69]. Briefly, metabolite extracts were dried, resuspended in pyridine, and derivatized with methoxylamine and N-tert-Butyl-dimethylsilyl-N-methyltrifluoroacetamide with 1% tert-Butyldimethylchlorosilane before GCMS analysis on an Agilent 7890 GC with Agielnt 5977 MS. Peak areas of all possible labeling states were extracted for full-carbon-backbone fragments of selected metabolites to obtain mass distribution vectors (MDVs). MDVs were then corrected for natural abundance of heavy isotopes by constrained optimization of the linear equation

$$I = L \cdot M,$$

where $I$ denotes the measured fractional abundance of metabolite fragments of different labeling states, $L$ denotes the correction matrix, and $M$ denotes the corrected MDV[69]. The fractional contribution from exogenous substrates was then calculated as the weighted average of the MDV

$$FC = \frac{\sum_{i=0}^{n} i \cdot s_i}{n},$$

where FC denotes the fractional contribution, $i$ is the position in the MDV, $s_i$ is the value of the MDV at the position $i$, and $n$ is the length of the MDV. All calculations were implemented in an in-house R script.

**Mitochondrial membrane potential and mtDNA/nDNA measurement**. Zebrafish cells or human cells were stained with tetramethylrhodamine, methyl ester (Image-iT™ TMRM Reagent, Invitrogen, ref. I34361; 50 nM) at 28 or 37 °C, respectively, for 30 min, then washed with PBS-5% FBS and analyzed by flow cytometry with LSRFortessa or LSR II cell analyzers (BD Biosciences). MFI was quantified using FlowJo software (Tree Star, Inc.). To measure the ratio of mitochondrial and nuclear DNA (mtDNA/nDNA), genomic DNA was extracted from $10^5$ cells using DNeasy Blood & Tissue Kit (Qiagen) and real-time qPCR was performed as described before[70]. Briefly, quantitative PCR reactions were assembled as follows: 2 μL of template DNA (3 ng/μL isolated DNA), 2 μL of mtDNA or nDNA target-specific primer pair (400 nM final concentration each), 12.5 μL SYBR Green PCR Master Mix, and 8.5 μL $H_2O$ in 1 well of the 96-well PCR plate. All qPCR experiments were performed in triplicate wells in three independent experiments. Primer sequences can be found in Supplementary Table 2.

**Reactive oxygen species**. ROS were determined by incubating cells with Mito-SOX™ Red Mitochondrial Superoxide Indicator (Invitrogen, ref. M36008; 5 μM) at 37 °C for 10 min in the dark, then washed with PBS-5% FBS and analyzed by flow cytometry with LSRFortessa or LSR II cell analyzers (BD Biosciences). MFI was quantified using FlowJo software (Tree Star, Inc.).

**Western blot**. Total cell lysates from all protein samples in this manuscript were incubated on ice in lysis buffer for 20 min (50 mM Tris, pH 7.5, 150 mM NaCl, 1% Triton X-100, 10% glycerol, 1 mM EDTA, 1× protease and phosphate inhibitors (Sigma-Aldrich)) followed by three cycles of 15 s sonication (Bioruptor). Lysates were subjected to Western blot analysis to detect Ty tag (1:500, SAB4800032 Sigma-Aldrich), Flag tag (1:5000, Sigma-Aldrich, clone M2, F1804), β-actin (1:10,000, C4, Santa Cruz Biotechnology, sc47778), HA tag (1:1000, Sigma-Aldrich, 3F10, 000000011867423001), H3 (1:10.000, Abcam, ab4729), PPARβ (1:500, Santa Cruz Biotechnology, F-10, sc-74517), total oxphos (1:2000, Antibody Cocktail, Abcam, ab110413), AMPKα (1:2000, Cell Signaling Technology, D5A2, 5831), and phospho-AMPKα T172 (1:2000, Cell Signaling Technology, 40H9, 2535). For detection of LC3B (1:2000, Cell Signaling Technology, 2775) the cells were additionally subjected to a 24 h treatment with chloroquine (Sigma-Aldrich, ref. C6628; 25 μM) before lysis. Western blots were either exposed on films or scanned with the BioRad ChemiDoc Touch Imaging System. Uncropped western blots can be found in Supplementary Fig. 8.

**Statistical analysis**. Statistical analysis was performed using two-tailed unpaired Student's $t$-test or one-way ANOVA test as indicated in the figure legends. Sample sizes and significance are shown in the figure legends. Statistical analysis for the overlap between gene sets was performed with hypergeometric test and shows the significance of the overlap in each case. Other statistical analyses are described in the respective sections.

**Genome-wide analysis**. RNA-Seq libraries, RNA-Seq preparation and analysis methodology, ATAC-Seq libraries, ATAC-Seq analysis methodology, Chromatin Immunoprecipitation, ChIP-Seq libraries, ChIP-Seq analysis, gene ontology, pathway and network analysis, motif analysis, digital genomic footprinting for ATAC-Seq, network construction, hierarchical differentiation tree can be found in the Supplementary Information.

**Code availability**. All computer codes used in this manuscript are available upon request.

**Data availability**. All raw sequencing data have been deposited in the Short Read Archive SRA under the BioProject accession codes PRJNA390228, PRJNA390119, and PRJNA433488. All the data are available without restrictions.

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

## Acknowledgements

We would like to thank the Deep Sequencing, Imaging, FACS, Fish, and bioinformatics facilities of the MPI for resources and support and David Stachura for valuable protocols. We thank colleagues for reading this manuscript. F.G.K. was supported by a return scholarship of the Forschungskommission, Faculty of Medicine, University of Freiburg, a grant of the Wissenschaftliche Gesellschaft in Freiburg (Scientific Society in Freiburg), and an EXCEL-Fellowship of the Faculty of Medicine, University of Freiburg, funded by the Else–Kröner–Fresenius–Stiftung. E.T. was supported by the Max Planck Society, a Marie Curie Career Integration Grant (631432 Bloody Signals) and the Deutsche Forschungsgemeinschaft, Research Training Group GRK2344 "MeInBio –BioInMe". E.T. and M.W.W. were supported by The Fritz Thyssen Stiftung (Az 10.17.1.026MN).

## Author contributions

I.P. and T.C. performed the bulk of experiments. R.I.K.G., N.Y., X.L., S.L., L.K., P.M., and J.D.C. performed experiments. C.C.A.B. and F.G.K. performed the CHT smear experiments in zebrafish. M.M. made the Tg(*UAS:HLX-GFP*) and Tg(*Mmu.Runx1:GAL4*) lines and gave useful advice. A.P. (mainly), D.v.E., and Alex.P. performed the bioinformatics analysis. P.C. performed the digital footprinting analyses. J.M.B. performed metabolomics and analysis. A.R., M.W.W., and E.L.P. provided reagents and critical insights. E.T. conceived that project and wrote the paper together with I.P. and input from all authors.

## Additional information

**Competing interests:** E.L.P. is a founder of Rheos and SAB Member of Immunomet. The remaining authors declare no competing interests.

