## [Peer Review File · Nature Communications]

Reviewers' comments:

Reviewer #1 (Remarks to the Author):

In this study, Piragyte et al elucidated the mechanisms of HLX function in normal hematopoiesis using zebrafish and human hematopoietic progenitor cells. The authors showed that HLX OE increases PPAR δ signaling, reduces mitochondrial membrane potential, and blocks myeloid cell maturation. HLX knockdown decreased the number of colony formation. The paper includes some potential interesting findings; however, many of key experiments have not been performed. PPAR δ signaling not only coordinates cellular metabolism but also has various functions, and there is no direct evidence showing that the roles of HLX-PPAR in hematopoiesis is mediated by the metabolic interplay. It is also not sufficient to address "HSC" maintenance. The paper may add some knowledge for HLX as a regulator of the metabolic cell state. However, this reviewer does not feel that it is new enough or is befitting for publication in this journal, and therefore believes that submission to a more specific journal should be appropriate.

Specific major comments:

1. Balance of HSC maintenance and differentiation:

The most experimental strategies assessed the differentiation of hematopoietic progenitor cells toward myeloid cells, but are not the one to compare the balance between the maintenance and differentiation. For example, to evaluate specification of HSC fate, Vannini et al isolated cells that divided one time and assessed their functions (Nat Commun, 2016), and these types of experiments are necessary. Authors should tone down this point or consider changing the title. In addition, CFU-C evaluates the differentiation capacity of progenitor cells. The roles of HLX in HSC maintenance must be addressed in the different assays (e.g. in vivo xenograft). The authors should modify the experimental strategies.

2. Cellular metabolisms:

As mentioned above, one of the weakness of this manuscript is a lack of the direct evidence for cellular metabolisms as the effector of HLX-PPAR. PPAR is well-known to be involved in the regulation of a variety of processes, and mitochondrial metabolism is just one of them. This is the key point, and the authors should provide solid data.

Mediators of oxidative phosphorylation were found in GSEA, and FAO genes showed changes in chromatin accessibility in *hlx1* MO (Fig. 2, 3). The essential roles of clearance of mitochondria by autophagy in HSCs and hematopoiesis have recently been reported by multiple studies (Ho et al., 2017, Vannini et al., 2016, Ito et al., 2016). PPAR-FAO in HSC stemness has been reported. Authors should assess if autophagy and/or FAO contribute in regulating hematopoiesis by HLX.

It is not clear that membrane potential is regulated by HLX-PPAR.

1) HLX overexpression or knockdown changes mitochondrial membrane potential, measured by TMRM (Fig. 4b, c). The membrane potential should reflect the base line OCR, though Seahorse data suggests that HLX mainly affects spare respiratory capacity (Fig. 4a). Additionally, the difference of baseline OCR is observed in human CD34+ cells (Fig. 5d). Authors should clearly explain how this discrepancy is coming from.

2) Statistical analysis (Vehicle v.s. GSK in Control) should be provided (Fig. 4d, e). Please explain why GSK3787 specifically increases mitochondrial mass in hHLX OE, but increases TMRM in both hHLX OE and Control (Fig. 4d). Authors should also discuss why GSK3787 showed no significant affect myeloid maturation in Control (Fig. 4e).

3. HLX leukemia:

As the contribution of HLX to AML has been firmly established (Kawahara M et al., 2012), it is interesting to ask whether the link between HLX and metabolism also exists in leukemic cells. Authors already used human leukemic cell lines in some experiments in this study. Authors should

explore if metabolic interplay by HLX contributes leukemogenesis. For example, HLX leukemic cells have low TMRM and high PPAR δ ? PPAR δ antagonist can inhibit leukemogenesis? These data will obviously increase the impact of this study.

4. Fig. 5:

Presented data in Fig. 5 are still immature.

a. Why # of CFU-C for scrRNA and GFP is so different (90 v.s. 40)? Why BFUE is so high in GFP (or HLX) than scrRNA? Authors should also double-check the label of the right panel is correct.

The proposed model (Fig. 6) suggests HLX OE leads to myeloid differentiation block. However, CFU-C data shows HLX OE increases the number of "mature" colonies (e.g. G, M colonies). Authors want to say that HLX OE promotes myeloid differentiation, while its differentiation is blocked? Please comment.

b. Which colony is M (or GEMM) in CD34+HLX?

e. Please comment on why L165,041 reduces TMRM of shHLX but not scrRNA.

f. There seems to be the effect of GSK3787 on CD34+ GFP. Authors should provide the statistical analysis between Vehicle and GSK3787 in CD34+ GFP.

h. Overall, why GEMM and GM colonies are few? Did authors confirm the purity of CD34+?

Other points

1. Fig. 3i:

The protein levels should be quantified.

Reviewer #2 (Remarks to the Author):

Manuscript by Piragyte et al., aims at uncovering the function of a non-clustered H2.0-like homeobox transcription factor (HLX) in hematopoiesis. HLX is known to be critical for hematopoietic stem cell maintenance. The authors have performed a large amount of work. Using gain of function and loss of function approaches in zebrafish they demonstrate the HLX is essential for hematopoietic stem cells; they also show that overexpression of HLX block myeloid differentiation. Both these results were already published several years ago in human cells (Kawahara et al., Cancer Cell 2012) and therefore not novel (although they confirm that the HLX function in hematopoiesis is conserved among species).

Additionally using mostly zebrafish but also human cell lines K562 and HL60 they conduct RNA-Seq and ChIP-Seq analyses. Using RNA-Seq they compare gene expression in overexpression or knockdown of HLX versus WT cells. These results identify oxidative phosphorylation and mitochondrial dysfunction as the most perturbed group of genes. They further validate the modulated expression of several metabolic/mitochondrial related genes specifically overexpression mildly affected the expression of electron transfer genes. In addition they find major changes in genes encoding for ribosomal proteins. They further use ATAC-Seq to measure chromatin accessibility. These experiments confirm mitochondrial dysfunction and OXPHOS as main genes affected by chromatin accessibility. They identify some of the fatty acid oxidation genes and PPAR δ with modulations of chromatin accessibility. In addition they find using GO and IPA analyses of the genes whose expression correlated with HLX in zebrafish that mitochondrial dysfunction and OXPHOS are among the pathways that are correlated with HLX expression in AML samples.

Next, they perform HLX ChIP-Seq in two mammalian lines K562 and HL-60; these experiments identified ETC and PPAR δ as direct targets of HLX. They also observed that perturbations of HLX expression are associated with modulations of mitochondrial membrane potential. They analyzed then the impact of PPAR δ pharmacological modulations on rescue of myeloid differentiation block

in zebrafish and these experiments led to the expected results.

These experiments overall led to identification of electron transfer chain and PPAR as one potential set of genes that may mediate HSC maintenance and differentiation defects. Since both PPAR and HLX are known regulators of HSC, the novelty of this manuscript remains unclear.

Finally they use human CD34+ cells and show that HLX induces a myeloid maturation block. This data confirms publication by Steidl's group in 2012 in Cancer Cell (Kawahara et al.). The authors are unable to make an overall mechanistic conclusion as to how metabolic changes are responsible for HSC myeloid differentiation block (and/or AML phenotype).

My additional comments are:

- The authors should show the mechanism by which the mitochondrial dysfunction/modulations of ETC are responsible for the HSC myeloid block.
- The authors should show how is the overexpression of HLX contributing to AML
- There are a number of over-interpretations: for instance page 9: TMRM is reduced in HLX OE is interpreted as "these data demonstrate that HLX regulates oxidative phosphorylation"

Reviewer #3 (Remarks to the Author):

Background:

In this manuscript, Piragyte et al. investigate the role of H2.0-like homeobox transcription factor (HLX) in hematopoiesis by modulating the expression of HLX in zebrafish, human hematopoietic cell lines and CD34+ progenitor cells. In zebrafish, the authors utilize both hHLX overexpression and *hlx1* knockdown systems to assess the role of HLX in hematopoietic differentiation. HLX overexpression resulted in expansion of the HSPC population while *hlx1* knockdown led to a depletion of the HSPC population. RNA-Seq and ATAC-Seq profiling was performed demonstrating that pathways involved in hematopoietic differentiation, mitochondrial dysfunction, and oxidative phosphorylation were dysregulated. ATAC-Seq and ChIP data revealed differences in chromatin accessibility and HLX binding genome-wide and at specific genes such as *ppar δ* , a gene involved in fatty acid oxidation and recently linked to LT-HSC stemness. The authors also aimed to assess the impact of HLX1 disruption in human HSPCs. HLX binds the PPAR locus in human hematopoietic cell lines in addition to other regions of active chromatin. Further mining of publically available data could be useful to characterize HLX expression changes during human and murine hematopoietic differentiation. HLX knockdown in hematopoietic cell lines led to decreased expression of PPAR and metabolic dysregulation, validating the role of HLX in regulation of hematopoietic differentiation via metabolic pathways. Functional assays utilizing both PPAR agonist and antagonists were able to nicely rescue the defects caused by HLX1 modulation. The authors use a diverse range of model systems and methods to clearly show that HLX1 disruption impacts hematopoietic differentiation, however, from this assessment, it remains unclear whether HLX1 is directly linked to regulating stemness. Moreover, while PPAR and metabolism pathways seem to be linked to HLX1 regulation, it remains unclear whether this is the direct pathway by which HLX1 may be impacting HSPC stemness.

Major comments:

(1) Most of the described data regarding the role of HLX in regulating stemness are indirect. The authors mention previous studies implicating HLX in LT-HSCs, but do not functionally address that in this study. Are there differences in replating ability of hematopoietic stem cells with KD or OE of

HLX? What is the expression of HLX in human and murine HSPC populations? Publically available datasets should be mined to assess the expression of HLX in different HSPC populations.

(2) Some of the data in Figure 5A regarding the role of HLX in human hematopoietic progenitor cells is unclear. KD of HLX results in depletion of all lineages, suggesting that HLX has an earlier role in regulating stemness. However, OE of HLX results in expansion of the erythroid population at the expense of the myeloid populations, suggesting a more myeloid-centric event.

(3) The mechanism of HLX gene regulation is not well established. The authors generate ATAC-Seq, RNA-Seq and ChIP-Seq data in zebrafish and human cell lines. These data should be overlaid to determine if they can drill down on a more clear set of genes that are dysregulated with modulation of HLX expression. What percentage of genes are negatively correlated in gain-of-function and loss-of-function experiments? Integration of these datasets would be mechanistically powerful. Also, functional validation of these expression changes of genes identified in previous figures should be validated in the CD34+ cells (i.e. PPAR and NDUF genes).

(4) These data have implications for the study of HLX in hematopoiesis, however, the role of HLX in human AML is unclear. Cancer databases (such as the cbiportal) show that the gene is almost never altered in AML and other malignancies. The therapeutic relevance of HLX has not yet been clearly delineated.

(5) The genome-wide analyses are spread across human and zebrafish (ChIP + histone mods only in humans, ATAC only in zebrafish), and it is not clear that results from one species are applicable to the other. What percentage of the differentially expressed gene/region pairs in zebrafish are well conserved in human?

(6) Overlap of HLX ChIP-Seq peaks with histone data in Figure 3b would be useful. Do these peaks map specifically to known enhancer sites defined by publically available ChIP-seq data?

(7) A more detailed exploration and discussion of the expression dynamics of HLX and progenitors would have been useful, is HLX expression restricted to HSCs and myeloid progenitors? There is now a significant amount of publically available data for human and mouse transcriptome and even epigenome profiling. The authors should use these data as a reference and include description (at the very least) of HLX expression across progenitors to better contextualize these findings. In addition the authors should also look to see if the HLX ChIP bound sites in the genome are known enhancers determined by ATAC-seq measurements in human hematopoietic progenitors (PMID: 27526324). Overall, the analysis may be significantly improved if only ATAC-seq peaks that vary in hematopoietic progenitors and are also de-regulated in AML are considered.

Minor comments:

(1) In main text figures showing HLX KD or OE, please show the level of HLX expression. For example, the westerns in Figure 3I and qPCR figures in Figure 2 and 3 should show HLX expression levels compared to wildtype. HLX expression levels in the zebrafish models should also be shown.

(2) It is unclear whether the PPAR agonist and antagonist are hitting their target. Demonstrating target inhibition or activation would be useful.

(3) The rescue in differentiation and plating in Figure 5f and 5h are not entirely convincing, however, HLX likely has varied transcriptional functions which would result in an incomplete rescue of activity. Also, it is perplexing that the early GMP/monoblasts seems to have an enhanced differentiation phenotype in Figure 5f but this does not impact the % of downstream mature lineages. Is there a change that reads out in absolute cell number of these lineages?

(4) Do the authors see changes of HOX accessibility at motif sites in HLX loss-of-function and gain-of-function studies, a classical phenotype in hematopoietic differentiation?

(5) How overexpressed is HLX in ChIP assays and does level of overexpression lead to non-specificity?

(6) CRISPR/Cas9 targeting on specific enhancers to functionally validate specific regulatory units would be interesting.

(7) Figure 2a shows differentially expressed genes relative to the control, but figure 2b has the control as another condition, raising the question of what it is being compared to.

(8) The methods section indicates that both MACS 1.4.2 and 2.1.0 were used for peak calling in the ATAC and ChIP analyses. The versions are known to produce different results from the same

input, but no statistical significance thresholds are provided, nor is it specified what each version was used for.

(9) The authors use IPA to look for enrichment of proximal genes to chromatin accessibility changes. The authors should also use GREAT, as this is the standard in the field for epigenomic analysis.

(10) The methods for the analysis of patient derived AML gene expression signatures was confusing. The main text should be updated to contain details as to what data was interrogated and how many patients were used.

Manuscript NCOMMS-17-17758-T. Piragyte *et al.*

Response to the main points made by the reviewers and editor:

Here is a summary of the new figures that we have added.

Figure 2d and e
Figure 4b (revised)
Figure 5 a to g
Figure 6 d,g
Figure 8 a to f
Sup. Figure 1a
Sup. Figure 2f
Sup. Figure 3a to c
Sup. Figure 4b (revised),e
Sup. Figure 5
Sup. Figure 6c

1) Demonstrate the role of HLX in HSC maintenance (Reviewer #1), if the focus of the manuscript on HSC maintenance is to be maintained.

As suggested by Reviewer #1, we have refocused the revised manuscript on myeloid differentiation and removed all references to HSC maintenance. We attempted to perform serial re-plating of CD34⁺-HLX and control cells, but our CD34⁺ cells were of insufficient quality to yield conclusive results. Therefore, we have not included these results in the revised manuscript. However, it may be interesting to note that, while control cells did not produce colonies upon replating, CD34⁺ HLX-overexpressing cells produced 30 colonies, suggesting that HLX is involved in HSC maintenance.

2) Both Reviewer #1 and #2 highlight the need to explore a potential link between HLX and metabolism in leukemic cells.

We thank Reviewers #1 and #2 for bringing this important point to our attention. To address this issue, we examined the expression levels of HLX in different AML lines and then performed several experiments (below) in THP1 cells (subtype M5), which have the highest levels of HLX and also exhibit high PPAR δ expression at the RNA and the protein level (new Figure 8a,b). To summarize these results:

First, we tried to check what happens when we overexpress HLX in an AML cell line.

(a) ChIP-seq for HLX in THP1 cells confirmed that HLX binds *ETC* genes and *PPAR δ* (new Figure 5a and Supplementary Table 4) in THP1 cells.

(b) H3K27ac ChIP revealed that HLX binds to open chromatin/active enhancers (see also response to Reviewer #3) (new Figure 5b and Supplementary Table 4).

(c) Analysis of ATAC-seq data from a previously published paper (PMID: 27526324), on HSCs, pre-leukemic HSCs, and leukemic HSCs, and comparison of these data with HLX peaks (our work), showed that the majority of HLX peaks fall in ATAC-seq peaks in all these samples (new Figure 5c).

(d) Analyses of metabolic parameters, such as OCR and ROS production, revealed similar effects on THP1 cells, when compared to normal cells overexpressing *HLX* (new Figure 5e,f).

(e) Measurements of carbon incorporation from glucose into metabolites shows that THP1 *HLX* overexpressing cells accelerate their fatty acid synthesis (new Sup. Figure 5), consistent with the involvement of PPAR δ in fatty acid metabolism.

(f) Western blots with AMPK and LC3-II antibodies showed that *HLX* overexpression leads to activation of AMPK and autophagy (Figure 5d).

(g) Finally, we measured TMRM and autophagy in a variety of AML lines (new Fig. 8c,d), but it was very hard to correlate all these parameters unless we do an in depth study.

Since we had established that PPAR δ can rescue the HLX associated phenotypes in normal cells, we checked whether PPAR δ inhibition can lead to lethality or terminal differentiation of AML cell lines. This was not the case (new Figure 8e). However, since we have established this new link of AMPK activation, we tried to use an AMPK inhibitor. Indeed, the AMPK inhibitor dorsomorphin could cause lethality in AML cells, but minimally affected normal cells (new Figure 8f).

All the experiments described above verify the fact that HLX controls

metabolic regulation also in leukemic cells. However, we have to highlight that many differences exist between normal and leukemic cells.

3) Strengthen the link between HLX-PPAR-metabolism-haematopoiesis, as suggested by Reviewers #1 and #3.

Following the suggestion of Reviewers #1 and #3, we further examined the role of HLX in regulating metabolism during hematopoiesis. As stated before, we do not believe that PPAR δ is the only metabolic parameter we should examine. Thus, our goal was to prove that, indeed, metabolism is involved in hematopoietic regulation mediated by HLX. **First**, we performed ChIP-seq for HLX in the AML cell line THP1. We consistently observe binding of HLX to PPAR δ in the same genomic regions (new Figure 5a). **Second**, we proved that HLX bound regions are located in H3K27ac enriched regions (new Figure 5b) and are also accessible in HSCs, preleukemic and leukemic HSCs (new Figure 5c) and thus could play a role in both physiological and pathological conditions. **Third**, we show robust upregulation of PPAR δ and downregulation of ETC genes, upon HLX overexpression (new Figure 5d). **Fourth**, we show that HLX-overexpressing cells have increased fatty acid synthesis (new Sup. Fig 5), consistent with the involvement of PPAR δ in fatty acid metabolism. **Fifth**, by screening AML cell lines, we show that THP1 cells express the highest levels of HLX and this is followed by high expression of PPAR δ (new Figure 8a). **Last**, we show that there is an important overlap between publicly available ChIP data for PPAR δ in myeloid cells and our list of genes deregulated in CD34⁺ cells upon HLX overexpression or knockdown (new Supplementary Table 6). This evidence, together with our previously shown rescue experiments by modulating PPAR δ establish the fact that metabolism is regulated by HLX. However, we do not claim either that PPAR δ is the only effector or that the rescue is exclusively based on metabolic changes.

4) In light of previous work, all reviewers emphasize the need to provide further mechanistic insight, in particular into how mitochondria/ETC alterations

could lead to the phenotype of HSCs or myeloid differentiation blockage.

We agree this is an important point, but it is also a very broad and complex question. It is established that specific metabolic profiles are essential for HSC identity, but also for differentiated lineages (PMID: 28504706, PMID: 28504707, PMID: 27738012, PMID: 27731316). For example, downregulation of autophagy causes accumulation of mitochondria, ultimately leading to accelerated differentiation (PMID:28241143). In our data we observe low mitochondrial activity and elevated autophagy levels both consistent with metabolic profiles of undifferentiated cells that are in agreement with a differentiation block. A significant number of studies also suggest that mitochondria and metabolism are crucial for leukemic transformation, for example, in AML (PMID: 27179622, PMID: 25640960 etc).

Although it is well established that metabolism is important for cell fate, we did try to obtain some direct evidence on how low mitochondrial activity affects myeloid differentiation. We first asked what could be the metabolic reaction of the cells to low mitochondrial activity. It is well established that mitochondria manipulation leads to AMPK activation. For example addition of rotenone, a known inhibitor of mitochondrial electron transport, is known to activate AMPK (PMID: 26816379, PMID: 29106036). We hypothesized that HLX-induced low *ETC* gene expression may also result in AMPK activation in our settings. Western blots with anti-AMPK α and anti-p-AMPK α (active AMPK) show that AMPK is indeed upregulated and activated in TPH1 cells (new Figure 5d). We next asked whether this combined effect of inhibition of the ETC chain and activation of AMPK can indeed affect myeloid differentiation. Indeed, treatment of human primary HSPCs with metformin, an AMPK activator that also blocks the mitochondria respiratory chain complex I, resulted in a myeloid maturation block, similar to the effects of overexpressing HLX in these cells (new Figure 6f). These results demonstrate that metabolic deregulation caused by *HLX* overexpression, can lead to the myeloid differentiation block, at least in part.

Point-by-point response to the reviewers' comments:

Reviewer #1:

In this study, Piragyte et al elucidated the mechanisms of HLX function in normal hematopoiesis using zebrafish and human hematopoietic progenitor cells. The authors showed that HLX OE increases PPAR δ signaling, reduces mitochondrial membrane potential, and blocks myeloid cell maturation. HLX knockdown decreased the number of colony formation. The paper includes some potential interesting findings; however, many of key experiments have not been performed. PPAR δ signaling not only coordinates cellular metabolism but also has various functions, and there is no direct evidence showing that the roles of HLX-PPAR in hematopoiesis is mediated by the metabolic interplay. It is also not sufficient to address "HSC" maintenance. The paper may add some knowledge for HLX as a regulator of the metabolic cell state. However, this reviewer does not feel that it is new enough or is befitting for publication in this journal, and therefore believes that submission to a more specific journal should be appropriate.

We thank the reviewer for the valid suggestions. We agree that our data provide insufficient evidence for a direct involvement of HLX in HSC stemness and we have therefore removed such claims in the revised manuscript. We also agree that PPAR δ , as well as HLX, have many other functions besides regulating metabolism, and we have emphasized this point in the discussion part of the revised manuscript. However, our already existing rescue experiments demonstrate that the myeloid maturation block caused by HLX overexpression is mediated, at least partially, by metabolic deregulation. **First, pharmacological inhibition of PPAR δ rescued the mitochondrial membrane potential phenotype and the myeloid differentiation block caused by HLX overexpression in zebrafish embryos and human CD34⁺ cells, and conversely, a PPAR δ agonist rescued HSPC loss in *both* systems. **Second,****

metformin, an AMPK activator, induced a myeloid differentiation block in CD34⁺ cells, suggesting that activation of the AMPK pathway affects myeloid differentiation (new Figure 6g). This experiment indicates that metabolic deregulation can directly affect myeloid differentiation. **Finally**, AMPK pharmacological inhibition with dorsomorphin reduced viability in all the AML cell lines tested, but did not significantly affect control myeloid progenitor cell populations (new Figure 8f). Together these results provide important novel mechanistic insights into the role of HLX and metabolism in promoting AML, and highlight the potential therapeutic application of using AMPK inhibitors to treat leukemia.

Specific major comments:

1. Balance of HSC maintenance and differentiation:

The most experimental strategies assessed the differentiation of hematopoietic progenitor cells toward myeloid cells, but are not the one to compare the balance between the maintenance and differentiation. For example, to evaluate specification of HSC fate, Vannini et al isolated cells that divided one time and assessed their functions (Nat Commun, 2016), and these types of experiments are necessary. Authors should tone down this point or consider changing the title. In addition, CFU-C evaluates the differentiation capacity of progenitor cells. The roles of HLX in HSC maintenance must be addressed in the different assays (e.g. in vivo xenograft). The authors should modify the experimental strategies.

We appreciate this valuable comment and, as suggested by the reviewer, we have now removed any reference to the possible function for HLX in stemness from the revised manuscript and we have also changed its title.

2. Cellular metabolisms:

As mentioned above, one of the weakness of this manuscript is a lack of the direct evidence for cellular metabolisms as the effector of HLX-PPAR. PPAR is well-known to be involved in the regulation of a variety of processes, and

mitochondrial metabolism is just one of them. This is the key point, and the authors should provide solid data.

We have followed Reviewer #1's suggestion and performed more experiments to further assess that metabolism is regulated by HLX and can be, at least partly, responsible for the hematopoietic phenotypes caused upon HLX perturbations. The revised manuscript now includes the following robust evidence:

Molecular evidence

- RNA-seq in zebrafish and in human primary CD34⁺ cells upon *HLX* knockdown and overexpression show that HLX regulates mitochondrial genes and PPAR δ . We only focused on these genes, but several more metabolic genes were found (new Figure 6 Supplementary Table 5).
- western blots show that PPAR δ protein is upregulated upon HLX overexpression (new Figure 5d).
- ChIP-seq for HLX in THP1 cells (M5 subtype AML cell line) demonstrates that HLX directly binds to metabolic genes, including *PPAR* and *ETC* genes (new Figure 5a and Supplementary Table 4).
- Using publicly available ChIP-seq data for PPAR δ , we show that many deregulated in CD34⁺ cells upon *HLX* overexpression or knockdown could be direct PPAR δ targets (new Supplementary Table 6). Many of these common genes regard metabolic regulation. We therefore believe that, at least partially, PPAR δ mediates its effects through metabolic regulation.

Functional evidence

- Additional metabolic analyses, such as OCR and ROS measurements (new Figure 5e,f), in THP1 show that *HLX* overexpression/knockdown affects metabolism.
- AMPK pathway (a metabolic sensor) is activated and autophagy is increased in *HLX* overexpressing cells (new Figure 5d,g).
- metformin, an AMPK activator that blocks complex I of the mitochondrial respiratory chain and thus mimics the metabolic function of HLX, causes a

myeloid differentiation block in CD34⁺ human cells (new Figure 6g). This is an important experiment, since it shows that metformin by itself can cause a myeloid block thus verifying metabolism as causative for the phenotype.

- glucose tracing shows increased fatty acid synthesis in *HLX* overexpressing cells, consistent with the literature showing that PPAR δ affects fatty acid metabolism (new Sup. Figure 4).

Collectively, these data show that HLX regulates directly the cellular metabolic status. Moreover, they suggest that metabolic deregulation caused by *HLX* overexpression/knockdown in HSCs may underpin the hematopoietic phenotypes, at least in part. We believe it is the panel of metabolic changes caused by *HLX* overexpression that results in the phenotype and not the overexpression of PPAR δ alone, as it is now clearly stated in our discussion. Thus, although there are likely other mechanisms at play during this process, in our study we have provided important new mechanistic insights into the role of HLX and metabolism in regulating myeloid differentiation.

Mediators of oxidative phosphorylation were found in GSEA, and FAO genes showed changes in chromatin accessibility in hlx1 MO (Fig. 2, 3). The essential roles of clearance of mitochondria by autophagy in HSCs and hematopoiesis have recently been reported by multiple studies (Ho et al., 2017, Vannini et al., 2016, Ito et al., 2016). PPAR-FAO in HSC stemness has been reported. Authors should assess if autophagy and/or FAO contribute in regulating hematopoiesis by HLX.

We thank the reviewer for these suggestions. As mentioned above, we have performed glucose tracing and found that *HLX* overexpressing cells have increased Fatty Acid Synthesis, which hints to the fact fatty acid metabolism is affected (new Supplementary Figure 5). We also tried to assess FAO by palmitate tracing. Unfortunately, our cells incorporated only traces of palmitate and thus we cannot draw valid conclusions.

In addition, we also assessed autophagy by western blot and showed

that, upon *HLX* overexpression, LC3 II protein levels are increased in THP1 cells, revealing that *HLX* increases autophagy rates (new Figure 5g). These results again indicate that *HLX* controls hematopoiesis through the regulation of metabolism.

It is not clear that membrane potential is regulated by HLX-PPAR.

1)HLX overexpression or knockdown changes mitochondrial membrane potential, measured by TMRM (Fig. 4b, c). The membrane potential should reflect the base line OCR, though Seahorse data suggests that HLX mainly affects spare respiratory capacity (Fig. 4a). Additionally, the difference of baseline OCR is observed in human CD34+ cells (Fig. 5d). Authors should clearly explain how this discrepancy is coming from.

We agree with the reviewer. However, there is variability in basal line OCR between our biological replicates, both in zebrafish and in CD34⁺ human cells. This variability may be due to compensatory mechanisms, or a failure to sustain constant levels of *HLX* overexpression. For this reason, we only considered robust and reproducible changes in maximal respiratory capacity. It is also established that spare respiratory capacity can reveal mitochondrial defects that can be compensated in steady state (PMID: 17611283, 21726199). In addition to this, the notion that basal OCR should reflect the staining with membrane potential dyes such as TMRE or TMRM is not universally the case, and in our experience is rarely a linear correlate, but this is beyond the scope of this current manuscript. The measurement of membrane potential, together with the pH (proton concentration) in the mitochondria is what is considered as the proton motive force, which affects the electron flow through the ETC complexes towards the ATP synthase (complex V). These factors together could influence the rate of oxygen consumption of complex IV, therefore to link oxygen consumption purely to the level of TMRM might be an oversimplification (PMID 21486251). For example, treatment of cells with the ATP synthase inhibitor oligomycin drops OCR, but increases the mitochondrial membrane potential. The inverse is true for treatment with FCCP, which drops TMRM staining drastically, but leads to

increased oxygen consumption with or without prior treatment with oligomycin. This shows that the staining of membrane potential is not correlated to the ability or regulation of oxygen consumption by mitochondrial complex V. Therefore the readout of maximal and/or spare respiratory capacity after blockade of complex V with Oligomycin, followed by treatment with FCCP (an ETC uncoupler) reveals true differences in the ETC ability to reestablish the proton gradient across the mitochondrial membrane, which as expected from the blots in Figures 3a, 5e and 6e is blunted by HLX expression.

2) Statistical analysis (Vehicle v.s. GSK in Control) should be provided (Fig. 4d, e).

We have included the statistical analyses requested by the reviewer in the revised manuscript.

Please explain why GSK3787 specifically increases mitochondrial mass in hHLX OE, but increases TMRM in both hHLX OE and Control (Fig. 4d).

Like is the case with OCR, mitochondrial mass and TMRM are not always linked and may be affected by several factors. For example, mitochondrial mass is affected by the biogenesis and the degradation of mitochondria whereas TMRM depends also on the substrate specificity and the ATP-synthase activity (PMID 21726199). Seeing as AMPK is upregulated in *HLX* overexpressing cells, and this is accompanied by increased autophagy, this could partly explain differences between the two cell types, as increased autophagy can enhance recycling of less functional mitochondria (PMID 28009285), which was linked to fate choices in T cells between self renewal versus terminal differentiation. Perhaps enhanced mitochondrial turnover and, the different metabolic status of control versus *HLX* OE cells could explain why GSK produced disparate effects.

Authors should also discuss why GSK3787 showed no significant affect

myeloid maturation in Control (Fig. 4e).

GSK3787 may indeed have an effect on myeloid maturation in control cells at higher concentrations. However, since higher concentrations of GSK3787 can convert cells into a glycolytic state, and would therefore affect differentiation, we chose to use lower concentrations in our experiments. In addition, the AMPK status of the cells might be a stronger determinant of the mitochondrial state, and the fate choices in the switch between self/renewal and terminal differentiation as shown in T cells (also see answer to the previous question). Whereas this is a very interesting question for a follow up study, we believe this is beyond the scope of the current study.

3. HLX leukemia:

As the contribution of HLX to AML has been firmly established (Kawahara M et al., 2012), it is interesting to ask whether the link between HLX and metabolism also exists in leukemic cells. Authors already used human leukemic cell lines in some experiments in this study. Authors should explore if metabolic interplay by HLX contributes leukemogenesis. For example, HLX leukemic cells have low TMRM and high PPAR δ ? PPAR δ antagonist can inhibit leukemogenesis? These data will obviously increase the impact of this study.

This is an excellent point and we are thankful to the reviewer for bringing it to our attention. As mentioned above, we performed several experiments in AML cell lines, particularly in THP1 cells (M5 subtype) (new Figure 5), including HLX ChIP-seq. THP1 cells have elevated HLX and PPAR δ mRNA and protein expression (new Figure 8 a,b). However, TMRM is variable between different leukemic cell lines and does not follow HLX expression (new Figure 8c). Of course many other factors contribute to TMRM and we cannot account for differences between cell lines, unless we thoroughly investigate their metabolic background. Importantly, although a PPAR antagonist did not affect AML cells (new Figure 8e), we provide new evidence that AMPK inhibition

caused lethality in the majority of AML cell lines used, without severely affecting normal CD34⁺ cells in myeloid differentiation media (new Figure 8g). This striking result may have promising therapeutic applications for the treatment of AML.

4. Fig. 5:

Presented data in Fig. 5 are still immature.

We have improved this figure in the revised manuscript.

a. Why # of CFU-C for scrRNA and GFP is so different (90 v.s. 40)? Why BFUE is so high in GFP (or HLX) than scrRNA? Authors should also double-check the label of the right panel is correct.

The number of CFU-C varies between scrRNA and GFP because the vector used for overexpressing *HLX* and *GFP* is an inducible lentiviral vector, which is a considerably larger vector. Regarding the BFU-E colonies, this was an error of mislabeling and we have corrected this mistake in the revised manuscript (BFUE were interchanged with Macrophages). We apologize for this labeling error and we thank the reviewer for pointing it out to us.

The proposed model (Fig. 6) suggests HLX OE leads to myeloid differentiation block. However, CFU-C data shows HLX OE increases the number of “mature” colonies (e.g. G, M colonies). Authors want to say that HLX OE promotes myeloid differentiation, while its differentiation is blocked? Please comment.

The reviewer is right. In the CFU-C assay (Fig. 6a) it seems that the mature myeloid colonies are more upon *HLX* overexpression and that could mean that *HLX* promotes myelopoiesis with a concomitant differentiation block. This could be consistent with our results in zebrafish where we observed a shift towards myelopoiesis in the expense of erythroid differentiation (data not

shown). However, when we examined closely myeloid differentiation of *HLX* overexpressing cell, we did not observe this phenomenon (Fig. 6c, 7a. Sup. Fig. 6b). Thus, although we cannot exclude this possibility, we did not verify truly this in our experiments.

b. Which colony is M (or GEMM) in CD34+HLX?

We have indicated the specific colonies in Figure 6b of the revised manuscript.

e. Please comment on why L165,041 reduces TMRM of shHLX but not scrRNA.

It is possible that the different metabolic state of *HLX*-knockdown cells makes them more sensitive to PPAR δ modulation than control cells at least in the concentrations used.

f. There seems to be the effect of GSK3787 on CD34+ GFP. Authors should provide the statistical analysis between Vehicle and GSK3787 in CD34+ GFP

This analysis has now been performed and is included in the revised manuscript (Figure 7a). This difference was not statistically significant.

h. Overall, why GEMM and GM colonies are few? Did authors confirm the purity of CD34+?

Yes, we confirmed the purity of our CD34+ cells as presented here.

Other points

1. Fig. 3i:

The protein levels should be quantified.

We have used new western blots where the changes in protein levels are obvious and we have also quantified those differences (new Fig. 5d,g new Fig. 8b).

Reviewer #2:

Manuscript by Piragyte et al., aims at uncovering the function of a non-clustered H2.0-like homeobox transcription factor (HLX) in hematopoiesis. HLX is known to be critical for hematopoietic stem cell maintenance. The authors have performed a large amount of work. Using gain of function and loss of function approaches in zebrafish they demonstrate the HLX is essential for hematopoietic stem cells; they also show that overexpression of HLX block myeloid differentiation. Both these results were already published several years ago in human cells (Kawahara et al., Cancer Cell 2012) and therefore not novel (although they confirm that the HLX function in hematopoiesis is conserved among species).

We thank the reviewer for recognizing our efforts. We are aware that it was previously shown by Kawahara et al that HLX has an effect on myeloid differentiation and we have mentioned this clearly in our manuscript. However, our study significantly extends beyond the findings of this paper. First of all, this study was entirely performed on murine cells. Thus, our study shows for the first time that HLX regulates myeloid differentiation in normal (i.e. non-malignant) human cells. Our study also includes zebrafish as a model thus proving evolutionary conservation of HLX function. Moreover, we provide important new insights into the molecular function of HLX as a regulator of cell metabolism, and we identify, for the first time in any system, the direct targets of HLX mediating this function. Finally, in the current version of the manuscript we extended our findings regarding mechanistic aspects of HLX functions and we included experiments in AML cell lines. Thus, our study uncovers a new function for HLX in metabolic regulation and provides important new insights into the mechanisms underlying its role in myeloid differentiation, with potential therapeutic applications for leukemia patients.

These experiments overall led to identification of electron transfer chain and PPARD as one potential set of genes that may mediate HSC maintenance and differentiation defects. Since both PPARD and HLX are known regulators of HSC, the novelty of this manuscript remains unclear.

We respectfully disagree with the reviewer on this point. The identification and functional validation of PPARD as a direct target of HLX is novel and significant, since it uncovers a new unexpected function for HLX in metabolic regulation. Together with additional evidence present in the current manuscript, our study provides a molecular mechanism for the role of HLX in myeloid differentiation. To our knowledge, our findings significantly increase our understanding of the function of this important but understudied transcription factor.

Finally they use human CD34+ cells and show that HLX induces a myeloid

maturation block. This data confirms publication by Steidl's group in 2012 in Cancer Cell (Kawahara et al.). The authors are unable to make an overall mechanistic conclusion as to how metabolic changes are responsible for HSC myeloid differentiation block (and/or AML phenotype).

As we mentioned above, the research performed by Steidl's group was entirely performed on mouse cells. Although murine models are extremely valuable, validation of findings in the mouse on human cells is fundamental (and common practice), particularly for functions implicated in human disease. Moreover, it is well established that metabolic changes are essential for HSC maintenance and differentiated blood lineages (PMID: 28504706, PMID: 28504707, PMID: 27738012, PMID: 27731316), but also for AML (PMID: 27179622, PMID: 25640960 etc), and therefore it was never our intention to repeat these investigations. The aim of our study was to identify direct targets of HLX mediating its function in hematopoiesis, in order to understand the mechanistic underpinnings of this important function of HLX, an understudied transcription factor.

Our data not only identified novel HLX effectors, but also uncovered a new role for HLX in metabolic regulation. Moreover, we show that this metabolic function of HLX partially (but not exclusively) accounts for the hematopoietic phenotypes of HLX, as you can see from our answer to your next question.

My additional comments are:

- *The authors should show the mechanism by which the mitochondrial dysfunction/modulations of ETC are responsible for the HSC myeloid block.*

This is an important point and we have addressed it in the revised manuscript. It is well established that downregulation of mitochondria function leads to elevated AMPK levels. For example rotenone, an inhibitor of mitochondria electron transport chain leads to AMPK activation (PMID: 26816379, PMID:

29106036). We then asked whether *HLX*-overexpressing cells that present downregulation of *ETC* genes expression have also activated AMPK, an important metabolic stress sensor. Indeed, we found that AMPK α and p-AMPK α (activated) protein levels are increased in *HLX*-overexpressing cells (new Figure 5d). To assess whether AMPK activation may block myeloid differentiation, we treated CD34⁺ human primary cells with metformin, an AMPK activator that also affects the mitochondrial complex I, thus mimicking the function of *HLX*. Indeed, we found that AMPK activation mimics the myeloid differentiation block phenotype caused by *HLX* (new Figure 6g). These data provide some further insights into how metabolic dysfunction can affect myeloid differentiation.

- *The authors should show how is the overexpression of *HLX* contributing to AML*

We followed the reviewer's suggestion, and examined the function of *HLX* in AML cell lines. We performed several experiments on THP1 cells (see also details in 'general response' above), which belong to M5 subtype leukemia, in which *HLX* is most overexpressed (new Fig 5). ChIP-seq revealed that *HLX* also binds to metabolic genes in these cells, including *PPAR δ* and *ETC* genes (new Fig. 5a and Sup. Table 4). Moreover, *HLX* overexpression in THP1 cells has similar effects on cell metabolism, when compared to zebrafish or primary human HSPCs (new Fig. 5e,f). Additionally, *HLX* overexpression increases the levels of *PPAR δ* , AMPK and autophagy in THP1 cells (new Figure 5d,g). Importantly, we show that even though *PPAR δ* inhibition did not affect the viability or differentiation of AML cell lines (new Fig. 8e), AMPK inhibition reduced the viability of all AML cell lines tested (new Figure 8f), with minimal effects on control cells. Our results support a previous report suggesting that AMPK activation protects leukemia-initiating cells from metabolic stress (PMID:26440282). Thus, this mechanism could be further investigated for potential therapeutic applications, but this is beyond the scope of our study.

- There are a number of over-interpretations: for instance page 9: TMRM is reduced in HLX OE is interpreted as “these data demonstrate that HLX regulates oxidative phosphorylation”

We have modified the text to improve clarity.

Reviewer #3:

The authors use a diverse range of model systems and methods to clearly show that HLX1 disruption impacts hematopoietic differentiation, however, from this assessment, it remains unclear whether HLX1 is directly linked to regulating stemness. Moreover, while PPAR and metabolism pathways seem to be linked to HLX1 regulation, it remains unclear whether this is the direct pathway by which HLX1 may be impacting HSPC stemness.

We agree with the reviewer and we have therefore removed all mentions of the possible role of HLX in HSC maintenance from the manuscript. However, we are convinced that we have provided sufficient evidence to show that metabolic regulation accounts, at least partially, for the hematopoietic phenotypes of HLX (see responses to the editor comments 3,4). Moreover, it is well established that metabolism is important for the identity of HSCs, but also differentiated lineages (PMID: 28504706, PMID: 28504707, PMID: 27738012, PMID: 27731316). For example, deletion of autophagy related genes leads to accumulation of mitochondria, ultimately resulting in accelerated differentiation (PMID:28241143). In agreement to our results, several studies also suggest that mitochondria and metabolism are crucial for leukemic transformation, including AML (PMID: 27179622, PMID). Our study uncovers a novel function for HLX in metabolic regulation and identifies direct metabolic targets, including in human primary and AML cells. Moreover, we provide functional data showing that this metabolic function of HLX accounts, at least partially, for the myeloid differentiation phenotypes (see also responses to the editor comments 3,4).

Major comments:

(1) Most of the described data regarding the role of HLX in regulating stemness are indirect. The authors mention previous studies implicating HLX in LT-HSCs, but do not functionally address that in this study. Are there differences in replating ability of hematopoietic stem cells with KD or OE of HLX? What is the expression of HLX in human and murine HSPC populations? Publically available datasets should be mined to assess the expression of HLX in different HSPC populations.

We agree with the reviewer and we have removed all mentions to HSC maintenance in the revised manuscript. We have mined publicly available datasets and included HLX expression profiles in the mouse and human hematopoietic systems in the revised manuscript (new Supplementary Figure 3a,b). We have also attempted to perform replating assays in CD34⁺ cells, but due to the low quality of our last batch of cells we were unsuccessful. Specifically, thrice the control cells did not form any colonies, but the HLX overexpressing cells produced approximately 30 colonies upon replating. However, as these experimental conditions are not suitable to produce reliable results, we have not included these results in the revised manuscript.

(2) Some of the data in Figure 5A regarding the role of HLX in human hematopoietic progenitor cells is unclear. KD of HLX results in depletion of all lineages, suggesting that HLX has an earlier role in regulating stemness. However, OE of HLX results in expansion of the erythroid population at the expense of the myeloid populations, suggesting a more myeloid-centric event.

We agree with the reviewer that knockdown experiments of HLX point to a general role of HLX in stemness. It is on our overexpression phenotypes that we did not address this. We mostly focused on myeloid differentiation, but that does not preclude other potential phenotypes regarding other lineages. However, we do not observe an expansion of the erythroid population, since this figure was mislabeled. We have corrected our mislabeling error in Figure

5a and it is now clear that there is no expansion of the erythroid population, but a possible expansion of myeloid populations in addition to the myeloid differentiation block. However, we did not confirm this expansion in other experiments (see also answer to question 4a or reviewer 1).

3) The mechanism of HLX gene regulation is not well established. The authors generate ATAC-Seq, RNA-Seq and ChIP-Seq data in zebrafish and human cell lines. These data should be overlaid to determine if they can drill down on a more clear set of genes that are dysregulated with modulation of HLX expression. What percentage of genes are negatively correlated in gain-of-function and loss-of-function experiments? Integration of these datasets would be mechanistically powerful. Also, functional validation of these expression changes of genes identified in previous figures should be validated in the CD34+ cells (i.e. PPAR and NDUF genes).

We thank the reviewer for the valuable suggestions. We have performed several comparisons to integrate our different datasets, and these results have been included in the revised manuscript (section: **a)** “HLX regulates genes involved in metabolism, **b)** “HLX directly regulates *ETC* and *PPAR* genes...” **c)** “*HLX* overexpression leads to elevated AMPK...”). Following the reviewer’s suggestion, we have also performed RNA-seq in CD34⁺ cells after knockdown or overexpression of *HLX* (new Fig. 6d and Sup. Table 5) and validated the expression changes of some of the identified genes by qPCR (new Sup. Fig. 6c). However, due to the diversity of these cells, we could not obtain statistical significance, and hence we only used fold-change as a cut-off for our analysis.

(4) These data have implications for the study of HLX in hematopoiesis, however, the role of HLX in human AML is unclear. Cancer databases (such as the cBioportal) show that the gene is almost never altered in AML and other malignancies. The therapeutic relevance of HLX has not yet been clearly delineated.

We have added a graph from cbioportal to the revised manuscript showing the HLX expression levels in AML patients and the predicted expression in equivalent normal cells (new Sup. Fig. 3c). Moreover, in Kawahara et al, 2012, the authors clearly mention that HLX was identified as an overexpressed gene in a significant portion of AML patients (PMID: 22897850).

(5) The genome-wide analyses are spread across human and zebrafish (ChIP + histone mods only in humans, ATAC only in zebrafish), and it is not clear that results from one species are applicable to the other. What percentage of the differentially expressed gene/region pairs in zebrafish are well conserved in human?

We have now included some of these dataset comparisons in the revised manuscript (section: “HLX directly regulates *ETC* and *PPAR* genes...”). Although it is challenging to compare human and zebrafish datasets because many genes are getting lost when we translate zebrafish genes to human orthologues, we were able to compare genes corresponding to ATAC-seq peaks in zebrafish with the genes identified by HLX ChIP in K562 and HL60 cells (in the above mentioned section of the text). Notably, 421 (19%) and 1431 (21%) bound genes in K562 and HL60 cells, respectively, also presented differential ATAC-seq peaks in zebrafish.

(6) Overlap of HLX ChIP-Seq peaks with histone data in Figure 3b would be useful. Do these peaks map specifically to known enhancer sites defined by publicly available ChIP-seq data?

Following the reviewer’s suggestion, we have included histone peaks from publicly available data in K562 cells in the revised manuscript (new Figure 4b). As previously shown in Figure 4f, we compared the HLX-bound regions in K562 cells with a panel of publicly available histone modification data. Interestingly, we found that there is a great enrichment for H3K4me1, which is

a marker for enhancers, suggesting that HLX peaks fall in enhancer regions. Similarly, when we also performed HLX- and H3K27ac-ChIP in THP1 cells and compared the bound regions, we found that the HLX peaks coincided with regions enriched for H3K27ac, which is a marker of active enhancers and promoters (new Fig. 5b).

(7) A more detailed exploration and discussion of the expression dynamics of HLX and progenitors would have been useful, is HLX expression restricted to HSCs and myeloid progenitors? There is now a significant amount of publically available data for human and mouse transcriptome and even epigenome profiling. The authors should use these data as a reference and include description (at the very least) of HLX expression across progenitors to better contextualize these findings.

We have included publicly available data for HLX expression in the revised manuscript (new Supplementary Fig. 3a,b). HLX is variably expressed in progenitor cells but also in myeloid lineages, but higher than normal HLX expression is associated with leukemia.

In addition the authors should also look to see if the HLX ChIP bound sites in the genome are known enhancers determined by ATAC-seq measurements in human hematopoietic progenitors (PMID: 27526324). Overall, the analysis may be significantly improved if only ATAC-seq peaks that vary in hematopoietic progenitors and are also de-regulated in AML are considered.

We thank the reviewer for this suggestion. We used sample data from this publication for normal HSCs, pre-leukemic HSCs, and leukemic HSCs. The majority of HLX peaks fall in ATAC-seq peaks from all of these categories (new Figure 5c). We did not attempt to include only ATAC-seq peaks that are deregulated between HSC and AML-HSCs because the AML-affected peaks vary from patient to patient, depending on the type of AML, for example.

Minor comments:

(1) In main text figures showing HLX KD or OE, please show the level of HLX expression. For example, the westerns in Figure 3I and qPCR figures in Figure 2 and 3 should show HLX expression levels compared to wildtype. HLX expression levels in the zebrafish models should also be shown.

A major technical challenge in our study was the lack of a specific HLX antibody. We tested several anti-HLX antibodies, but none gave reproducible results in human cells. Thus, we avoided performing western blots on the endogenous protein and used mostly qPCR in both zebrafish and human cells to measure HLX expression levels. We have now included one qPCR that shows the levels of HLX in zebrafish after knockdown and overexpression (new Sup. Fig. 1a corresponds to the WISH experiments but also to the qPCR experiments in Fig. 2h). In the previous Fig. 3h that is now 4h, this is not a qPCR analysis, but a representation of selected data from the RNA-seq and *HLX* levels can be found in the corresponding table. We have now included the levels of *HLX* overexpression in our western blots (new Fig. 5d)

(2) It is unclear whether the PPAR δ agonist and antagonist are hitting their target. Demonstrating target inhibition or activation would be useful.

The validity of the PPAR δ agonists and antagonists used in our study has been extensively and repeatedly verified in previous publications. For example, GSK3787 PMID: 20128594, PMID: 20516370 and L165,041 PMID: 25934804, PMID: 10037770, PMID: 17167170, PMID: 10818235. We have attempted to perform ChIP-seq for PPAR δ with inhibitors and activators, however, the antibody used previously for similar experiments (PMID: 25934804) is not produced anymore, and the newer PPAR δ antibody produced by the same company (Santa Cruz) did not work in our hands.

(3) The rescue in differentiation and plating in Figure 5f and 5h are not entirely

convincing, however, HLX likely has varied transcriptional functions which would result in an incomplete rescue of activity. Also, it is perplexing that the early GMP/monoblasts seems to have an enhanced differentiation phenotype in Figure 5f but this does not impact the % of downstream mature lineages. Is there a change that reads out in absolute cell number of these lineages?

We agree with the reviewer that a partial rescue was the expected outcome of this experiment, since HLX has pleiotropic functions. However, we repeatedly obtained an almost complete rescue of the myeloid differentiation block, whereas the rescue of the *HLX* knockdown was only partial. We carefully looked at all our experiments to see if by inducing a myeloid differentiation block we observe lower percentages of mature cells. This is true in zebrafish (Fig. 3e albeit not statistically significant and Fig 7a which is statistically significant). We are happy to provide absolute numbers to the reviewer, but since for every sample, different total numbers of cells are counted we can draw conclusions only from the percentages among viable cells.

(4) Do the authors see changes of HOX accessibility at motif sites in HLX loss-of-function and gain-of-function studies, a classical phenotype in hematopoietic differentiation?

To address this question, we examined the ATAC-seq data from wild type and *hlx1* MO zebrafish. By performing digital genomic footprinting from ATAC-seq data, we showed that Hoxc9 and Hoxa2 motifs were more accessible in the WT specific peaks compared to the *hlx1* MO specific peaks (new Figure 2d). Thus, *hlx1* MO appears to have lost peaks containing HOX motifs, which, as the reviewer said, is typical in hematopoietic differentiation.

(5) How overexpressed is HLX in ChIP assays and does level of overexpression lead to non-specificity?

We thank the reviewer for bringing this important point to our attention. *HLX* overexpression could indeed result in non-specific binding, and without a specific antibody it is difficult to rule this out in our experiments. However, we overcame this technical hurdle by inserting a 3xTy tag in the endogenous locus of *HLX* in K562 cells (heterozygous), and we also performed CHIP-PCR to verify the binding to *PPAR δ* and some other targets (Figure 4d). Additionally, we obtained similar results in 3 different systems, in zebrafish and humans, and validated some of the genes with functional experiments. Thus, our results are robust and we are confident that *HLX* overexpression specifically affects the genes identified in our experiments. Finally, since *HLX* is overexpressed in cancer our results have validity for this setting.

(6) CRISPR/Cas9 targeting on specific enhancers to functionally validate specific regulatory units would be interesting.

Since we do not know the exact base pairs that *HLX* binds to, we decided to do a transient experiment and delete one of the *HLX* bound genomic regions in proximity to *ATP11b* gene. Even in a transient manner, we obtained almost 90% deletion. qPCR for *ATP11b* gene expression showed upregulation of this gene upon deletion of the above mentioned regions (new Sup. Fig. 4e). Additionally, with all comparisons that we did with histone modifications and ATAC-seq peaks we believe that *HLX* bound regions are functional (Fig 4b,f and new Fig. 5b,c).

(7) Figure 2a shows differentially expressed genes relative to the control, but figure 2b has the control as another condition, raising the question of what it is being compared to.

We apologize for the labeling error in this figure. This is not Log fold change as stated but the z-transformed normalized gene expression values (Figure 2a). This has been corrected in the revised manuscript.

(8) The methods section indicates that both MACS 1.4.2 and 2.1.0 were used for peak calling in the ATAC and ChIP analyses. The versions are known to produce different results from the same input, but no statistical significance thresholds are provided, nor is it specified what each version was used for.

We have corrected this part of the material and methods section of the revised manuscript.

(9) The authors use IPA to look for enrichment of proximal genes to chromatin accessibility changes. The authors should also use GREAT, as this is the standard in the field for epigenomic analysis.

We have added GREAT analysis for the ChIP- and the ATC-seq data that we generated in the revised manuscript (new Sup. Fig. 2f, Sup. Fig 4b, new Fig. 5a and Sup. Tables 2 and 4).

(10) The methods for the analysis of patient derived AML gene expression signatures was confusing. The main text should be updated to contain details as to what data was interrogated and how many patients were used.

We have modified the text accordingly to improve the clarity of the manuscript.

Reviewers' comments:

Reviewer #1 (Remarks to the Author):

The authors have addressed my concerns.

Reviewer #2 (Remarks to the Author):

The authors are commended for their pursuit of a direct mechanism involving HLX regulation of metabolism in HSC differentiation. They revise the manuscript to focus on the function of HLX in HSC myeloid differentiation and abandon HLX involvement in HSC maintenance. They conduct a series of new experiments to establish the relationship between HLX and metabolism in the regulation of HSC differentiation. They provide a plethora of data and identify HLX binding to electron transfer chain genes and PPAR δ as direct targets in AML line THP1. What is disappointing is that the authors do not reach any clear conclusion beyond HLX regulates metabolism in normal and leukemic cells. One outstanding question is the identity of the direct target (s) of HLX that lead to mitochondrial dysfunction. Another one is whether HLX regulation of metabolism in leukemia cells differs from that of normal cells. A major problem with the experiments described here is the lack of a clear conclusion. A large number of studies published since 2008 suggest that mitochondrial metabolism is critical for HSC differentiation (references in Page 4 of the rebuttal are missing some important ones). In addition it is known that PPAR δ is essential for HSC maintenance. Many experiments described here establish an association without a direct evidence.

There are also a number of ambiguities. For instance, they conclude a number of times that they are unable to interpret their data or to make a coherent conclusion: Page 2(g) we measured TMRM and autophagy in a variety of AML lines....but it was very hard to correlate all these parameters unless we do an in depth study. Page 3: "However, we have to highlight that many differences exist between normal and leukemic cells"; page 11 last paragraph:.....Of course many other factors contribute to TMRM and we can not account for differences between cell lines, unless we thoroughly investigate their metabolic background....

The authors should also note that their repeated assertion that Steidl group's work published in 2012 is "entirely" in murine cells is incorrect.

Reviewer #3 (Remarks to the Author):

The authors largely attempted to address my comments and the current version of the manuscript is much improved. However, there are a few points I don't think were adequately addressed. Below is a summary of these remaining concerns:

1) In general, the new figure elements are convincing and more clearly bridges the gap between the genomic analyses and the proposed HLX function. However, there are a number of occasions where statistical significance was unclear or missing. Further attention to the statistical significance, especially when overlapping gene or gene-peak sets, would improve the quality of the analyses. As a general comment the authors should carefully review the statistical approaches in the manuscript and/or more clearly state significance in the main text. A few examples:

Figure 2c is great, however I had to dig into the supplementary methods to find how differential peaks were called. It'd be useful to write the significance threshold used in the main text, as is done for the RNA-seq analysis.

31% of deregulated genes showed change in chromatin accessibility, is this statistically significant?

406 and 399 deregulated genes upon HLX overexpression or knockdown may be directly bound by PPAR, is this overlap statistically significant?

2) The footprint analysis presented in Figure 2d is confusing as presented. As shown it is quite unclear how the authors derive gain of AP1/T-box and loss of HOX motifs. This analysis can be significantly improved by: i) more clear description of what has been done in the main text. ii) The two figures are independently clustered, thus they are difficult to compare visually. Preserving the order of the motifs across rows would be clearer. iii) Consider a more simple representation, how about a simple bar plot with example footprints for the AP1 and HOX motifs. At the moment it is unclear what the values represent and what changes are statistically significant, from my perspective this analysis may arise from just noise. iv) I see how HOX motifs are lost, however, many other factors appear to be lost as well, is this significant for the interpretation of the results?

Overall, I strongly suggest the authors to revisit and significantly improve this analysis, or replace it with a more intuitive aggregate footprint plot or similar.

3) The data presented in Figure 6f appears to be very weakly significant. Can the authors speak to the magnitude of this difference as it appears to be a very small change. The authors should consider repeating this experiment to see if it remains statistically significant.

Reviewer #4 (Remarks to the Author):

In their manuscript entitled "A metabolic interplay coordinated by HLX regulates myeloid differentiation", Piragyte et al. combine elegant experiments using zebrafish and human CD34+ HSCs to show the role of HLX in the biology of HSCs and myeloid differentiation. While it is already known that HLX plays a role in human hematopoiesis, they show that this role has been conserved through evolution.

There are still a few points that remain to be addressed concerning the work performed in the zebrafish model:

Figure 1a: the authors claim that the overexpression of hHLX induces the expansion of HSCs, after they look at *runx1* at 36hpf. At this stage, it would be better to talk about HSC specification and not about expansion. If the authors want to show expansion, they should look at the marker *cmyb* at 72 or 96 hpf in the caudal hematopoietic tissue, where HSCs undergo expansion. Moreover in this figure, it is not clear what the graphs represent... is that a direct quantification of the WISH signal or the result of a quantitative PCR?

Figure 1b: same comments concerning the quantification results. Moreover, the differences observed are quite low and could be due to variability. As an example, the authors claim that there is no difference in Supplemental Fig 1e when they compare the levels of expression of *efnb2a* between *hlx*-morphants and control embryos. These appear more different than the images shown on Figure 1a and 1b.

One question remains unanswered: is the role of HLX cell or non-cell autonomous in the zebrafish model? The authors present data showing that endothelial cells proliferate more in hHLX OE embryos. They also performed RNAseq on endothelial cells sorted from hHLX-OE or *hlx*-MO embryos. Of note, when sorting *fli1*:mCherry cells, the authors not only sort endothelial cells but also hematopoietic progenitors. Why not sorting as well *kdrl*:eGFP positive cells from hHLX OE embryos? In the end on page 7, they conclude that *hlx1* regulates the expression of ETC and *ppar*

genes, but the authors do not conclude on the cell types affected by this regulation.

One important point reside in the fact that *hlx* overexpression augments myelopoiesis. To prove their point, the authors score *pu.1* expression at 48hpf. The authors are probably aware that at this stage, HSCs have hardly emerged from the hemogenic endothelium and are just starting to colonize the CHT. Therefore, there is little chance that they have started to differentiate into myeloid cells. It would be therefore important to show that their effect is HSC-dependent or not, as other lineages could be involved in this, such as primitive myeloid cells or erythro-myeloid progenitors. If the authors want to show a link to HSCs, they should at least score myelopoiesis after 4 dpf.

Finally, they show on figure 1b that the effect of hHLX OE is myeloid specific. To make such a claim involves that you would at least show that other lineages are not affected. They already show that the number of T cell progenitors is augmented in the thymus rudiments. What about erythroid progenitors? Neutrophils? Another way to show that myelopoiesis is blocked would be to show that terminal markers for myeloid lineages are reduced (*mpx* for neutrophils, *mfap4* or *mpeg1* for macrophages, *cpa5* for mast cells)

Minor points:

Figure 1e is called before figure 1d... the authors should change the order of the panels in this figure.

Figure 3f: the authors claim that they can rescue the HSC loss observed in *hlx*-Morphants by adding a *ppar* agonist. It would be nice to see embryo imaging as well as WISH data to support this claim, rather than just a graph.

Response to reviewers' comments

New Figures:

Figure 1e

Figure 2d

Figure 2g (third graph)

Figure 6f (replacement)

Sup. Figure 1c, 1d, 1e

Sup. Figure 2d (second panel)

Sup. Figure 3b, 3d

Reviewer #1 (Remarks to the Author):

The authors have addressed my concerns.

We thank the reviewer for his help in improving our manuscript and we are happy to have addressed all the concerns raised.

Reviewer #2 (Remarks to the Author):

The authors are commended for their pursuit of a direct mechanism involving HLX regulation of metabolism in HSC differentiation. They revise the manuscript to focus on the function of HLX in HSC myeloid differentiation and abandon HLX involvement in HSC maintenance. They conduct a series of new experiments to establish the relationship between HLX and metabolism in the regulation of HSC differentiation. They provide a plethora of data and identify HLX binding to electron transfer chain genes and PPAR δ as direct targets in AML line THP1. What is disappointing is that the authors do not reach any clear conclusion beyond HLX regulates metabolism in normal and leukemic cells.

We respectfully disagree with the reviewer. In this manuscript:

- We have established the hematopoietic phenotypes of HLX knockdown and overexpression in zebrafish and human cells.

- We have shown that HLX directly regulates the mitochondrial electron transport chain genes and PPAR δ . We have shown that in zebrafish and different types of human cells.
 - We have shown that the function of HLX is evolutionary conserved.
 - We have shown that oxygen consumption rate and mitochondrial membrane potential are regulated by HLX.
 - We have shown that modulation of PPAR δ can rescue the hematopoietic effects of HLX deregulation on normal hematopoiesis. In addition, we showed that, by mimicking the metabolic effects of HLX overexpression with metformin, we can actually cause the same phenotype on myeloid differentiation.
 - We have shown that AMPK inhibition causes lethality to AML cell lines.
- All these findings create a mechanistic cascade from the upstream regulator (HLX) to the downstream direct targets (ETC and PPAR δ genes), to functional verification with metabolic assays and finally to rescue experiments.

One outstanding question is the identity of the direct target (s) of HLX that lead to mitochondrial dysfunction.

Chromatin immunoprecipitation is the golden standard for identifying direct targets of a transcription factor.

- We have shown by ChIP-seq in three different cell lines that HLX binds directly to ETC genes and PPAR δ (Figure 4b, 4c, 4d and 5a). We have included tables with all the HLX targets that were identified by ChIP-seq (Table 4).
- We have correlated HLX bound genes with gene expression changes upon HLX knockdown or overexpression (Figure 4g, 4h, Sup. Figure 5g and text). Specifically for ETC genes and PPAR δ we proved, both by genome-wide data but also with ChIP-qPCR and qPCR, that targets bound by HLX are also differentially expressed upon HLX modulation. This is proof that these genes are directly regulated.
- We have further verified this result at the protein level (Figure 5d).

- We have shown that by deleting an HLX bound genomic region using CRISPR approach, we affect the expression of the corresponding gene (Sup. Figure 5e).

Based on all of this evidence we have identified the direct targets of HLX.

Another one is whether HLX regulation of metabolism in leukemia cells differs from that of normal cells.

To this extent we have proven that:

- HLX overexpression leads to upregulation of PPAR δ and downregulation of ETC genes in normal (zebrafish and primary human cells) and malignant cells (Figure 2g, 4h, 5d and 6d). This is followed by metabolic changes in oxygen consumption rate and mitochondrial membrane potential (Figures 3a, 3b, 3c, 3d, 5e, 6e and 7c).
- PPAR δ modulation rescues the hematopoietic phenotypes of both HLX knockdown or overexpression in physiological conditions, but not in leukemic cells (Figure 3e, 3f, 7a, 7b, 8e).
- AMPK is upregulated upon HLX overexpression. Blockade of AMPK minimally affects normal cells, but is lethal for leukemic cells (Figure 8f).

A major problem with the experiments described here is the lack of a clear conclusion.

We have answered this comment on our first answer to Reviewer 2.

A large number of studies published since 2008 suggest that mitochondrial metabolism is critical for HSC differentiation (references in Page 4 of the rebuttal are missing some important ones). In addition it is known that PPAR δ is essential for HSC maintenance. Many experiments described here establish an association without a direct evidence.

We agree with the reviewer that mitochondrial metabolism is known to affect

HSC differentiation and we clearly state that in our manuscript. We have also clearly mentioned that previous publications have established PPAR δ as a regulator of HSC maintenance. Our work identifies an upstream regulator of mitochondria and PPAR δ in a detailed molecular cascade that, together with functional and rescue experiments, provide direct evidence for the role of HLX in myeloid differentiation.

References on the rebuttal letter are only exemplary citations and the full reference lists exist in the manuscript.

There are also a number of ambiguities. For instance, they conclude a number of times that they are unable to interpret their data or to make a coherent conclusion: Page 2(g) we measured TMRM and autophagy in a variety of AML lines....but it was very hard to correlate all these parameters unless we do an in depth study. Page 3: “However, we have to highlight that many differences exist between normal and leukemic cells”; page 11 last paragraph:.....Of course many other factors contribute to TMRM and we can not account for differences between cell lines, unless we thoroughly investigate their metabolic background....

The clear conclusions are that PPAR δ can rescue the hematopoietic phenotypes in normal but not malignant cells. AMPK inhibition causes lethality in leukemic cell lines. These conclusions are clearly stated in the manuscript.

The authors should also note that their repeated assertion that Steidl group’s work published in 2012 is “entirely” in murine cells is incorrect.

The reviewer is correct since there are some experiments in the Steidl paper that were done in leukemic cell lines. Our comment was referring to the primary human cells that were used in our study.

Reviewer #3 (Remarks to the Author):

The authors largely attempted to address my comments and the current version of the manuscript is much improved. However, there are a few points I don't think were adequately addressed. Below is a summary of these remaining concerns:

We thank the reviewer for his/her positive comments and we are happy to answer to the remaining concerns.

1) In general, the new figure elements are convincing and more clearly bridges the gap between the genomic analyses and the proposed HLX function. However, there are a number of occasions where statistical significance was unclear or missing. Further attention to the statistical significance, especially when overlapping gene or gene-peak sets, would improve the quality of the analyses. As a general comment the authors should carefully review the statistical approaches in the manuscript and/or more clearly state significance in the main text. A few examples:

Figure 2c is great, however I had to dig into the supplementary methods to find how differential peaks were called. It'd be useful to write the significance threshold used in the main text, as is done for the RNA-seq analysis.

We have included and highlighted this information in the manuscript.

31% of deregulated genes showed change in chromatin accessibility, is this statistically significant?

406 and 399 deregulated genes upon HLX overexpression or knockdown may be directly bound by PPAR, is this overlap statistically significant?

We performed hypergeometric test for all the overlaps between datasets and included only significant overlaps in the manuscript. All the significance thresholds are indicated and highlighted in the manuscript.

2) The footprint analysis presented in Figure 2d is confusing as presented. As

shown it is quite unclear how the authors derive gain of AP1/T-box and loss of HOX motifs. This analysis can be significantly improved by: i) more clear description of what has been done in the main text.

Due to clarity, we have decided to move Figure 2d to the Supplementary Figure 3c and replace it with a simpler representation in the main Figure 2.

These motifs originate from motif discovery results in specific footprint populations, as well as from relevant TFs involved in the regulation of the endothelial-to-haematopoietic transition (Choi et al. Cell Rep 2012). We have added a more clear description to the text and this change is highlighted.

ii) The two figures are independently clustered, thus they are difficult to compare visually. Preserving the order of the motifs across rows would be clearer.

We agree with the reviewer that it would be easier to preserve the order of motifs, but these heatmaps (now moved to the supplementary material) provide double information. The color intensity represents the z-score, which demonstrates the enrichment of self- and co- occurrences in the tested population over an evenly sized, random sampling (1000 iterations) from all footprints in the reciprocal population. It is computed as $Z=(x-\mu)/\sigma$ whereby x is the observed number of self- and co-occurrences per motif in the tested population, μ and σ the mean and standard deviation of the computed background self- and co-occurrences. On the other hand, the clustering shows motif co-occurrence. That is why the clustering is different between the two heatmaps.

iii) Consider a more simple representation, how about a simple bar plot with example footprints for the AP1 and HOX motifs. At the moment it is unclear what the values represent and what changes are statistically significant, from my perspective this analysis may arise from just noise.

We have replaced Figure 2d with average normalized Tn5 insertion profiles

around footprinted motifs in merged ATAC peaks and barplots that represent the relative occurrences of footprinted Hoxc9 and AP-1 motifs that are easier to understand as the reviewer points out correctly (Figure 2d and Sup. Figure 3d).

The footprinting profiles themselves show occupancy of Hoxc9 motifs in WT zebrafish, and decreased occupancy at those sites in HLX-morphant fish. AP-1 motifs show the converse pattern. These motifs were detected as enriched in WT- and HLX morphant- specific footprints. Footprints themselves correspond to regions where ATAC insertions are significantly lower than the surrounding regions, adjusted with an FDR of 0.01 as described in Piper et al. *Nucleic Acids Res* 2013.

The barplots (Sup. Fig. 3d) showing the relative occurrences of footprinted Hoxc9 and AP-1 motifs were obtained by dividing the number of footprints containing Hoxc9 or AP-1 motifs (where relevant) by the total number of footprints in WT-specific, WT/HLX-MO-shared, and HLX-MO-only footprints.

Regarding the statistical significance of footprinted motif self- and co-occurrences (now Supplementary Figure 3c): because we are testing for under- and over -enrichment, the significance thresholds of the computed z-scores corresponds to those of a two-tailed hypothesis; the critical values for a significance level of 0.05 is thus $|z| > 1.96$. Examination of the z-scores of enrichments of Hox motifs revealed that Hoxc9 shows significant enrichment over background in the WT vs MO comparison ($z=3.583120411$). Additionally, the AP1 motif shows significant enrichment over background in the MO vs WT comparison ($z=13.24101684$).

Additionally, we performed t-tests based on the footprinting occupancy scores at WT-specific Hoxc9 motifs using the WT and MO ATAC data, and the same at MO-specific AP-1 motifs. Footprinting occupancy scores were retrieved using a custom script of the pyDNase package, `dnase_fos_scorer.py`, described in Bevington et al. *EMBO* 2016, using the `-A` switch for ATAC data. Footprinting occupancy scores (FOS) between two experiments are directly comparable as they are computed using the following formula:

$$\text{FOS} = (\text{C}+1)/\text{L} + (\text{C}+1)/\text{R}$$

Whereby C is the average number of reads over the candidate region and L,R the average number of reads left, right of the candidate region, respectively (Neph et al. Nature 2012). FOSs were first tested for normality to make sure the t-test can apply using a Shapiro-Wilks test in R. As the resulting p-values indicated that FOSs did not follow a normal distribution, we performed a log2 transform, which resulted in normal data; following are the p-values for the Shapiro-Wilks normality test ($p \geq 0.05$ implies the data is normal)

Hoxc9 WT-specific, log2 WT footprinting occupancy scores, **p= 0.5599** (was 6.077e-05 without log2 transform)

Hoxc9 WT-specific, log2 MO footprinting occupancy scores, **p=0.38** (was 3.639e-06 without log2 transform)

AP-1 MO-specific, log2 WT footprinting occupancy scores, **p= 0.3597** (was 4.695e-10 without log2 transform)

AP-1 MO-specific, log2 MO footprinting occupancy scores, **p= 0.1146** (was 8.907e-11 without log2 transform)

We thus performed two-tailed, paired t-tests in R as each value corresponds to the same chromosomal region, and not assuming any direction in the relationship between both samples.

Following are the t-test p-values between FOSs at Hoxc9 and AP-1 motifs between WT and HLX-MO ATAC datasets:

Hoxc9 WT-specific, log2 WT vs MO footprinting occupancy scores, **p= 2.9884E-34** (was 8.64471E-31 without log2 transform)

AP-1 MO-specific, log2 WT vs MO footprinting occupancy scores, **p= 1.24262E-37** (was 7.34244E-35 without log2 transform).

Thus, our manuscript currently includes only these highly significant differences.

iv) I see how HOX motifs are lost, however, many other factors appear to be lost as well, is this significant for the interpretation of the results?

As it is obvious from all the genome-wide experiments in our work, HLX is a multifaceted transcription factor. We have focused on the metabolic aspect, but other changes occur. The lost factors are not significant for the interpretation of the results in this study, but they could be useful for a future study.

Overall, I strongly suggest the authors to revisit and significantly improve this analysis, or replace it with a more intuitive aggregate footprint plot or similar.

As mentioned above, we have replaced Figure 2d with simpler representations and moved this Figure to the Supplementary Figures.

3) The data presented in Figure 6f appears to be very weakly significant. Can the authors speak to the magnitude of this difference as it appears to be a very small change. The authors should consider repeating this experiment to see if it remains statistically significant.

We agree with the reviewer that the change is small and we have now performed six (three additional) independent experiments to show that the change is highly significant and repetitive (Figure 6f). We believe that the relatively small change may be attributed to the modest overexpression of HLX in these primary cells.

Reviewer #4 (Remarks to the Author):

In their manuscript entitled “A metabolic interplay coordinated by HLX regulates myeloid differentiation”, Piragyte et al. combine elegant experiments using zebrafish and human CD34+ HSCs to show the role of HLX in the biology of HSCs and myeloid differentiation. While it is already known that

HLX plays a role in human hematopoiesis, they show that this role has been conserved through evolution.

We thank the reviewer for appreciating our work. The role of HLX has been established in mouse hematopoiesis, but in humans it has only been associated with Acute Myeloid Leukemia (AML) and not physiological hematopoiesis. Additionally, the molecular mechanism of HLX function remained unknown.

There are still a few points that remain to be addressed concerning the work performed in the zebrafish model:

Figure 1a: the authors claim that the overexpression of hHLX induces the expansion of HSCs, after they look at *runx1* at 36hpf. At this stage, it would be better to talk about HSC specification and not about expansion. If the authors want to show expansion, they should look at the marker *cmyb* at 72 or 96 hpf in the caudal hematopoietic tissue, where HSCs undergo expansion.

We agree with the reviewer that at 36hpf we can talk about HSC specification and not expansion. For this experiment we overexpressed HLX using the *fli1a* regulatory element. However, in this system HLX overexpression is not sustained. Thus, this is not the best system to use and perform *c-myb in situ* hybridization in, at later time points.

To adequately answer this question, we used the *Tg(Mmu.Runx1:GAL4)* (created by Marina Mione using the regulatory elements described in Tamplin et al, Cell 2015) crossed with *Tg:(UAS:hHLX)* and drove HLX expression specifically in HSPCs. We performed *runx1* WISH at 26hpf and showed increased staining in the HLX overexpressing embryos compared to wild type (Sup. Figure 1c). We then performed *c-myb* WISH at 3dpf and qPCR at 5dpf and showed also a modest increase (Sup. Figure 1c and 1d). With this experiment we conclude that HSPC specification is increased in HLX overexpressing embryos and this increase is sustained.

Moreover in this figure, it is not clear what the graphs represent... is that a direct quantification of the WISH signal or the result of a quantitative PCR?

The graph represents a direct quantification of the WISH signal by imaging. We have clarified this on the figure and figure legend.

Figure 1b: same comments concerning the quantification results. Moreover, the differences observed are quite low and could be due to variability. As an example, the authors claim that there is no difference in Supplemental Fig 1e when they compare the levels of expression of *efnb2a* between *hlx*-morphants and control embryos. These appear more different than the images shown on Figure 1a and 1b.

We have clarified the quantification results in the figure and figure legend.

The differences observed in Figure 1a and 1b are significant as proven by quantification with imaging, which, to our knowledge, is the best possible quantification method for *in situ* experiments. These differences are also verified in another system:

- Regarding the morphant phenotype, in Figure 1g, we have used a fluorescent line and quantified the amount of HSPCs in the morphants. This result is significant and proves, in an independent manner, that HSPC numbers are lower in *hlx1* morphants.
- Regarding the overexpression phenotype: 1) we have used *pu.1* as an early marker for myelopoiesis and we quantified this result (Figure 1b). 2) We have counted the amount of immature myeloid cells and it is significantly higher in *fli:hHLX* overexpression fish, thus proving our point in an independent manner (Figure 1c). 3) In the present revised manuscript, we have also included qPCR and WISH for mature myeloid markers in fish that overexpress HLX only in HSPCs (*Runx:hHLXOE*) and shown that these markers are greatly diminished upon HLX overexpression (Sup. Figure 1d, 1e). 4) Using the same system

(Runx:hHLXOE), we observed significant differences in the amount of mature and immature myeloid cells upon HLX overexpression in HSPCs (Fig. 1e).

All the above-mentioned results proved that our conclusions in Figure 1a and 1b are valid and not due to variability.

One question remains unanswered: is the role of HLX cell or non-cell autonomous in the zebrafish model?

We agree with the reviewer that this point needed clarification. To this end, we opted for the verification of the HLX overexpression phenotype when HLX is specifically expressed in hemogenic endothelium and HSPCs using the construct described above: (*Tg(Mmu.Runx1:GAL4)* crossed to *Tg:(UAS:hHLX)*). Indeed, specific HLX overexpression in HSPCs leads to increased specification of HPSCs, as shown by *runx1* and *c-myb* staining (Sup. Fig. 1c) followed by a myeloid differentiation block (Fig. 1e and Sup. Fig. 1d, 1e). This experiment points to an intrinsic role of HLX in hematopoietic cells for the described overexpression phenotype.

The authors present data showing that endothelial cells proliferate more in hHLX OE embryos. They also performed RNAseq on endothelial cells sorted from hHLX-OE or *hlx*-MO embryos. Of note, when sorting *fli1:mCherry* cells, the authors not only sort endothelial cells but also hematopoietic progenitors. Why not sorting as well *kdr1:eGFP* positive cells from hHLX OE embryos?

Unfortunately, we need to create another fish line to be able to perform this sort in *fli:hHLXOE* embryos. Since we have functional experiments that prove the implication of HLX in metabolism and we verify these in human primary hematopoietic cells, we feel that our conclusions are adequately supported.

In the end on page 7, they conclude that *hlx1* regulates the expression of ETC and *ppar* genes, but the authors do not conclude on the cell types affected by

this regulation.

Our existing RNA-seq data indicated that this regulation is true in both endothelial cells and HSPCs. To prove that this, indeed, happens in HSPCs we overexpressed HLX specifically in HSPCs as described above and performed qPCR for *ETC* and *ppar* genes. Even though these qPCRs were performed in whole embryos, any difference in expression should stem from the function of HLX in HSPCs. These experiments verify that *Hlx1* regulates *ETC* and *ppar* in HSPCs (Fig. 2g and Sup. Fig. 2d). Together with our previous results, we can claim that HLX overexpression affects the expression of these genes in both endothelial cells and HSPCs.

One important point reside in the fact that *hlx* overexpression augments myelopoiesis. To prove their point, the authors score *pu.1* expression at 48hpf. The authors are probably aware that at this stage, HSCs have hardly emerged from the hemogenic endothelium and are just starting to colonize the CHT. Therefore, there is little chance that they have started to differentiate into myeloid cells. It would be therefore important to show that their effect is HSC-dependent or not, as other lineages could be involved in this, such as primitive myeloid cells or erythro-myeloid progenitors. If the authors want to show a link to HSCs, they should at least score myelopoiesis after 4 dpf.

The reviewer is right and this is a very valid point. To answer this question we used the *Tg: Runx-Gal4;UAS-HLX-GFP* described above. In this system, any difference in myeloid differentiation originates exclusively from HLX overexpression in HSPCs. Using this system we measured the percentage of immature, mature and erythroid cells at 5dpf upon HLX overexpression. This experiment showed that HLX overexpression in HSPCs leads to a myeloid differentiation block with significantly more immature and less mature myeloid cells (Figure 1e).

Finally, they show on figure 1b that the effect of hHLX OE is myeloid specific.

To make such a claim involves that you would at least show that other lineages are not affected. They already show that the number of T cell progenitors is augmented in the thymus rudiments. What about erythroid progenitors? Neutrophils? Another way to show that myelopoiesis is blocked would be to show that terminal markers for myeloid lineages are reduced (mpx for neutrophils, mfap4 or mpeg1 for macrophages, cpa5 for mast cells)

We are sorry that we gave the impression that the effect of hHLX OE is only myeloid specific. We focused on myeloid differentiation because HLX is implicated in Acute Myeloid Leukemia and we wanted to discover its mechanistic function in this lineage. In this manuscript, we are not focusing on possible effects on other hematopoietic populations. However, erythrocyte counts shows that the cell number in this compartment is not significantly changed upon HLX overexpression (Fig. 1c, 1e). This was also verified by WISH and qPCR for gata1 (Sup. Fig. 1d, 1e). Additionally, we performed WISH and qPCR for differentiated myeloid markers, as the reviewer suggested in Tg:Runx-Gal4;UAS-HLX-GFP. These experiments nicely showed that the myeloid differentiation is blocked (Sup. Fig. 1d, 1e).

Minor points:

Figure 1e is called before figure 1d... the authors should change the order of the panels in this figure.

We changed the order of these figures.

Figure 3f: the authors claim that they can rescue the HSC loss observed in hlx-Morphants by adding a ppar agonist. It would be nice to see embryo imaging as well as WISH data to support this claim, rather than just a graph.

Unfortunately, we didn't take a picture of these embryos. We used FACS sorting to quantify the differences, since we consider this a more accurate method for quantification.

REVIEWERS' COMMENTS:

Reviewer #3 (Remarks to the Author):

The authors have significantly improved the clarity of the statistical results and analyses, and have addressed all my concerns.

Reviewer #4 (Remarks to the Author):

The authors have adequately answered my requests.

Response to reviewer comments

REVIEWERS' COMMENTS:

Reviewer #3 (Remarks to the Author):

The authors have significantly improved the clarity of the statistical results and analyses, and have addressed all my concerns.

We thank the reviewer for his/her help.

Reviewer #4 (Remarks to the Author):

The authors have adequately answered my requests

We thank the reviewer for his/her help.